

# Modelling long-term alluvial peatland dynamics in temperate river floodplains

Ward Swinnen[1,2], Nils Broothaerts[1], Gert Verstraeten[1]

[1]Department of Earth and Environmental Science, KU Leuven, Leuven, 3000, Belgium
[2]Research Foundation – Flanders (FWO), Brussels, 1000, Belgium

*Correspondence to*: Ward Swinnen (ward.swinnen@kuleuven.be)

## Abstract

Peat growth is a frequent phenomenon in European river valleys. The presence of peat in the floodplain stratigraphy makes them hotspots of carbon storage. The long-term dynamics of alluvial peatlands are complex due to interactions between the
peat and the local river network, and as a result, alluvial peatland development in relation to both regional and local conditions is not well understood. In this study, a new modelling framework is presented to simulate long-term peatland development in river floodplains by coupling a river basin hydrology model (STREAM) with a local peat growth model (modified version of Digibog). The model is applied to two lowland rivers in northern Belgium, located in the European loess (Dijle river) and sand (Grote Nete river) belts. Parameter sensitivity analysis and scenario analysis are used to study the relative importance of
internal processes and environmental conditions on peatland development. The simulation results demonstrate that the peat thickness is largely determined by the spacing and mobility of the local river channel(s) rather than by channel characteristics or peat properties. In contrast, changes in regional conditions such as climate and land cover across the upstream river basin showed to influence the river hydrograph, but have a limited effect on peat growth. These results demonstrate that alluvial peatland development is strongly determined by the geomorphic boundary conditions set by the river network and as such
models must account for river channel dynamics to adequately simulate peatland development trajectories in valley environments.

## 1 Introduction

In many river systems throughout Europe, peat can be found in the Holocene floodplain stratigraphy (Notebaert and Verstraeten, 2010). The development of peat and organic-rich floodplain deposits is associated with low-energy floodplain
environments and limited sediment dynamics (Candel et al., 2017). Active peat growth occurred mostly during the early Holocene and has disappeared at many locations due to anthropogenic land cover change since the Neolithic and its effect on the landscape sediment dynamics (Broothaerts et al., 2014b; Swinnen et al., 2020). While these have been studied as environmental archives, the dynamics of alluvial peat formation and its interaction with the local geomorphology, sediment dynamics, hydrology and human impact are not fully understood (e.g. Comas et al., 2004). Most available case-studies on



floodplain peatlands cluster in tropical and boreal regions where human impact only occurred in more recent time periods and mainly involves geomorphic and stratigraphic reconstructions (e.g. Boucher et al., 2006; Householder et al., 2012; Kumaran et al., 2016; Mann et al., 2010).

Nevertheless, temperate river floodplains can store large amounts of organic carbon and contribute disproportionally to the total terrestrial carbon stock, often due to the presence of peat (Sutfin et al., 2016; Wohl et al., 2012). In recent times, floodplain

environments are increasingly used as multi-functional systems, providing a range of services such as flood retention, nature conservation and carbon storage (Brown et al., 2018). As such, a good understanding of the dynamics of alluvial peatlands in interaction with the complex nature of floodplain environments is crucial, not only to protect both active and buried peat but also to balance the different functions and services of a floodplain under changing climatic and anthropogenic conditions (Brown et al., 2018; van Diggelen et al., 2006; Notebaert and Verstraeten, 2010). Process-based peatland modelling studies on

these environments are not available and can be a useful tool in studying the interaction between peatland development and the environment, both on a local and on a river basin scale. A peatland model, specifically designed for an alluvial setting would allow to study these environments under a range of conditions, which is more difficult using only the traditional stratigraphic and multi-proxy reconstructions. For other peatland types such as peat bogs and blanket peatlands, the modelling of their long-term dynamics has demonstrated the added value of process-based models in improving our understanding of

peatland development and its sensitivity to both internal and external factors (Frolking et al., 2010; Morris et al., 2011; Yu et al., 2001a).

Several local peat growth models have been developed over the past decades with varying degrees of complexity and focus mostly on raised bogs or general peat processes (Frolking et al., 2010; Hilbert et al., 2000; Morris et al., 2012). A frequently used model type is the cohort model in which a peatland is represented by a peat column where each layer or cohort corresponds

to the biomass deposited in a specific year (Baird et al., 2012; Frolking et al., 2010; Heinemeyer et al., 2010). This model structure allows to simulate peat growth at an annual resolution over millennial timescales and allows to track and update specific peat properties trough time. One of the most recent and widely applied cohort models is the Digibog model, which incorporates a variety of hydrological and ecohydrological feedbacks (Morris et al., 2011). The model has been frequently used to study peatland behaviour both in temperate and tropical environments (Baird et al., 2017; Kelly et al., 2014; Young et

al., 2017, 2019). However, these models are not directly applicable to an alluvial setting. The assumption can be made that most of the basic concepts of raised bog development are also valid for alluvial peatlands, but even then, several changes will have to be made to these models to make them representative of an alluvial context. Firstly, most peatland models assume the peat column to be ombrotrophic and as such do not simulate water fluxes between the surrounding landscape and the peat layer. In river floodplains, this assumption cannot be made because of the hydrological interaction of the peat layer with the

river channel(s) and groundwater. In contrast to several other peatland types, the water fluxes feeding alluvial peatlands can be diverse and include both local and regional sources. As the water fluxes interacting with the alluvial peatland can originate from the entire upstream river basin, all factors influencing the river basin hydrology such as climate and land cover can potentially influence alluvial peatland dynamics. Secondly, the peat-forming environments in an alluvial context show a wide





variety of vegetation types, which can range from a combination of mosses and sedges to carr forests, which is less the case

for raised bogs.

Alluvial peatlands and the local river network can be assumed to co-evolve and should thus be considered as a peatland-stream complex. However, research on peatland streams has focussed mainly on their role in the carbon cycle, but understanding their geomorphology and channel morphodynamics have been more speculative (Billett et al., 2004; Candel et al., 2017; Juutinen et al., 2013; Watters and Stanley, 2007). Several conceptual models describing the development of the stream network in peat-

forming floodplains have been put forward, but currently, both the number of available case-studies and our understanding of the channel dynamics in alluvial peatlands is limited (Broothaerts et al., 2013; Candel et al., 2017; Lespez et al., 2015a; Nanson, 2009). Given the combination of both local and regional factors, influencing alluvial peatland dynamics, the development of a modelling framework requires a combination of a local peat growth model incorporating the local river network characteristics and a regional hydrology model.

Here, we present a coupled alluvial peatland – river basin hydrology model which is able to simulate alluvial peatland dynamics over Holocene timescales. The alluvial peatland model is based on the 1D Digibog model that has been adapted to floodplain settings, whereas the STREAM model is selected as a regional hydrology model due to its simplicity but still being able to simulate the river basin water balance under various conditions (Aerts et al., 1999; Morris et al., 2011). To simulate the river basin hydrology under both past and current conditions, an equilibrium must be found between the low data availability for

past conditions and the flexibility required to simulate the hydrology in a spatially explicit way. The model has been successfully applied to river basins across the globe, simulating palaeohydrology and river basin water balances (Aerts et al., 2006; Notebaert et al., 2011; Renssen et al., 2007; Ward et al., 2008). We apply the coupled model to two contrasting river basins in northern Belgium, located in the loess (Dijle river) and sand belts (Grote Nete river). These two cases are selected because of their contrasting hydrology, well-documented floodplain stratigraphy and landscape history and can serve as

representative case studies for the European loess and sand belts covering large parts of the European lowlands. Given the complexity of the studied environment and the limited available data and knowledge on alluvial peatlands, a scenario-based approach is used here, varying climatic conditions, land cover and river channel properties, which allows to explore the effect of different processes on the long-term development of alluvial peatlands. This methodology can provide insight in the sensitivity of these systems to changes in external factors and allows to identify key processes governing the evolution of these

environments.

## 2 Study area and field data

The study area consists of two river basins located in northern Belgium (fig. 1). The Dijle river basin (742 km²) is situated at the northern edge of the European loess belt and drains an undulating plateau with elevations ranging between 25 and 165 metres above sea level. The geology consists of Palaeogene and Neogene clays and sands, overlain by Pleistocene loess

(Herbosch and Verniers, 2013). The soils in the floodplain consist mainly of fluvisols and gleysols, while the hillslope are





covered by luvisols (Dondeyne et al., 2014). The current land use is dominated by arable land (41%) grassland (18%) and built-up land (26%). The Holocene geomorphic dynamics of the Dijle river have been studied in detail using both modelling and field-based techniques (De Brue and Verstraeten, 2014; Notebaert et al., 2009; Van Oost et al., 2012). Palynological studies in the floodplains of the Dijle show that the river basin was largely forested during the first half of the Holocene (Mullenders

et al., 1966; Mullenders and Gullentops, 1957; De Smedt, 1973). Significant indications of human impact start from the Bronze age onwards (3.9 ka cal BP), with a gradual opening of the vegetation and an increase in the area under cropland during the Roman period (Broothaerts et al., 2014c). The Dijle river has known widespread peat growth during the early Holocene, but almost no active peat growth today. The floodplain deposits are generally thick, with the organic-rich layers mostly covered by overbank sedimentation from the Roman period onwards following soil erosion on the deforested hillslopes (Broothaerts

et al., 2014a; Swinnen et al., 2020). The case of the Dijle river can be used as a model for many river systems in temperate climates, where floodplain peat deposits were covered by mineral sediment due to anthropogenic land cover change and increased erosion rates (Notebaert and Verstraeten, 2010; Treat et al., 2019; Walter and Merritts, 2008).

The Grote Nete river basin (561 km²) is situated at the southern side of the European sand belt. The river basin drains the Campine plateau, consisting of middle Pleistocene fluvial sands and gravels (Beerten et al., 2017; Gullentops et al., 2001). The

most frequent soil types consist of Arenosols, Podzols and Plaggic Anthrosols (Dondeyne et al., 2014). The current land cover consists mainly of grassland (48%), forest (21%) and built-up land (27%). Traces of agriculture in the region date back to 5.45 ka cal BP, but farming is assumed to have been limited to a few isolated locations until the Middle Ages due to the relatively infertile soils (Goedseels and Vanhautte, 1983; Goossens and Riksen, 2009). The Holocene floodplain stratigraphy of the Grote Nete river is relatively thin with a mean thickness around 1.5 metres, and highly variable, with alternating layers of channel

deposits, peat and overbank deposition (Swinnen et al., 2020). Peat cutting for household fuel has been documented for the region but the effect of this activity on the alluvial stratigraphy has not yet been quantified (Burny, 1999).



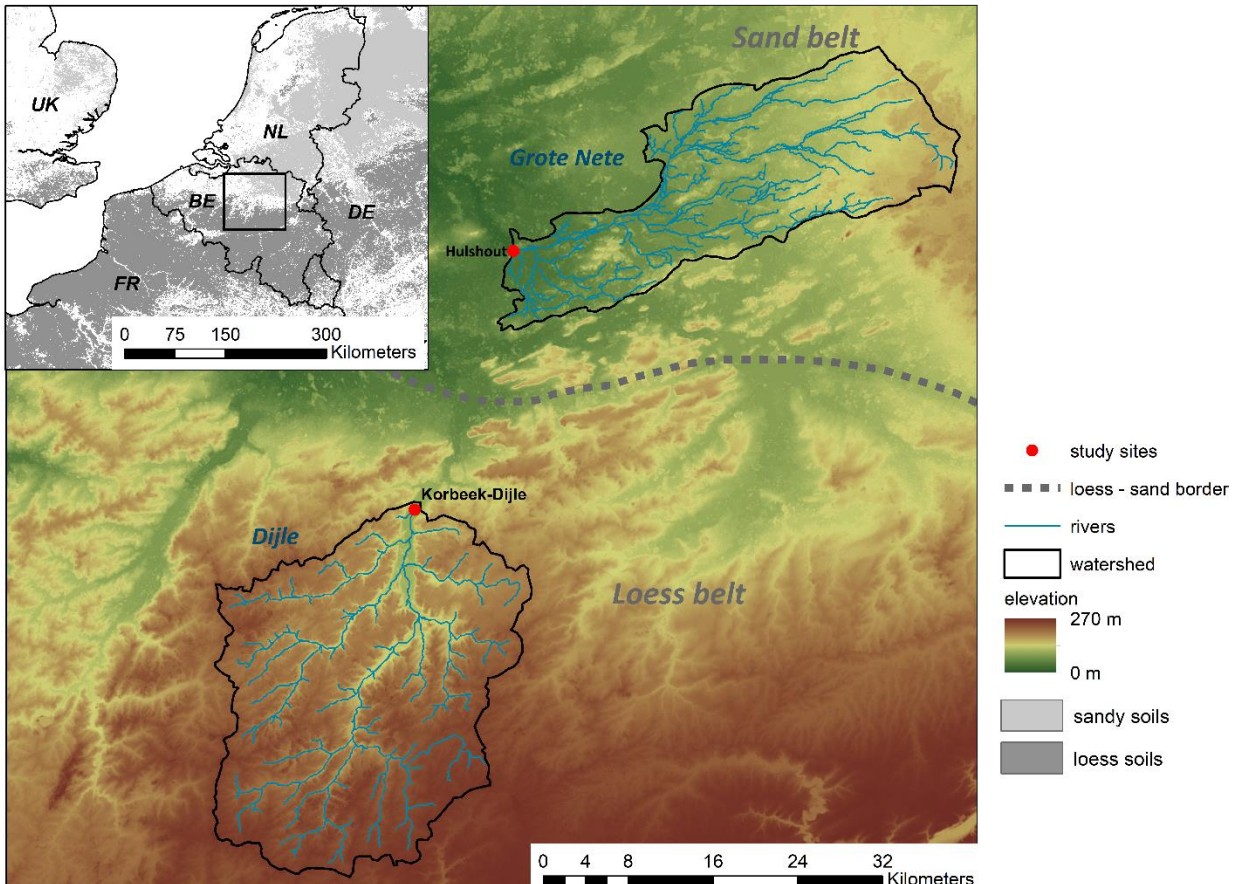

**Figure 1: Location of the two studied river basins (Dijle and Grote Nete), with indications of the floodplain study sites and the border between the sand and loess belts. The mapping of the sandy and loess soils is derived from the Topsoil physical properties for Europe-dataset (ESDAC) (Ballabio et al., 2016).**

Available field data for the two river basins can be used to constrain specific model parameters and serve as external validation of the model simulations. A dataset on peat thickness and properties, derived from soil corings across the study areas provides a mean dry bulk density for the alluvial peat deposits in the loess and sand belts (Swinnen et al., 2020). The measured peat thickness and bulk density values are corrected for compaction due to burial by mineral sediment. The compaction percentage is calculated using an empirical relationship, expressing the percentage of thickness reduction as a function of the effective stress of the overlying sediment (Van Asselen et al., 2010; Swinnen, 2020). Details on the calculation of the compaction-corrected peat thickness and dry bulk density values are given in the appendix, section A3. The mean and range in the observed compaction-corrected peat thickness values are used as a reference for the results of the sensitivity analysis and scenario simulations. As such, they can be used to identify which model scenarios correspond with the observed peat thickness. The compaction-corrected dry bulk density data are used to set the dry bulk density model parameter value to the measured mean value.



## 3 Model development

To simulate the development of alluvial peatlands at Holocene timescales, a modelling framework is developed by coupling a local peat growth model with a river basin hydrology model (fig. 2). The local model domain consists of a floodplain cross-

section with a flat impermeable substrate layer on top of which peat can grow. The peat layer is assumed to develop as an elliptic bog, located between straight adjacent channels. The local peatland model consists of a modified version of the Digibog peatland model which simulates a 1D peat profile at the centre of the bog. The entire floodplain width is assumed to be covered by peat, with the river channel(s) spaced equally over the cross-section (fig. 3). The cross-section locations are assumed to be situated at the outlet of the studied river basins, which corresponds to the localities of Korbeek-Dijle (Dijle river) and Hulshout

(Grote Nete river) (fig. 1). The water level in the river channels is simulated by the STREAM basin hydrology model, and serves as a lateral boundary condition for the local peat growth model (fig. 2).

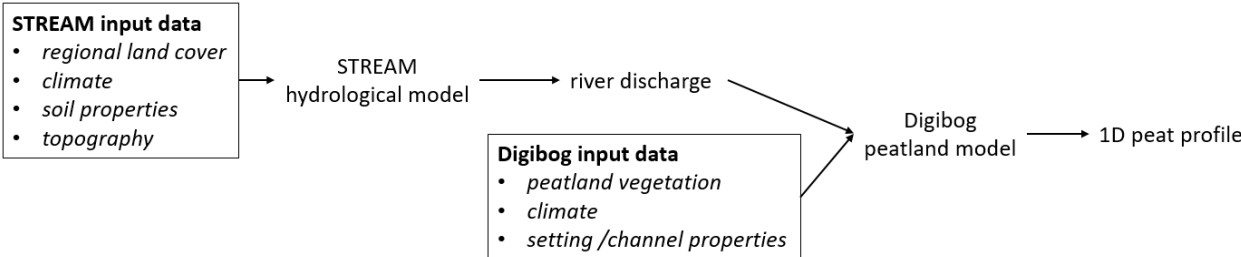

**Figure 2: General model workflow. For a more detailed description of the model structure, the reader is referred to the text.**

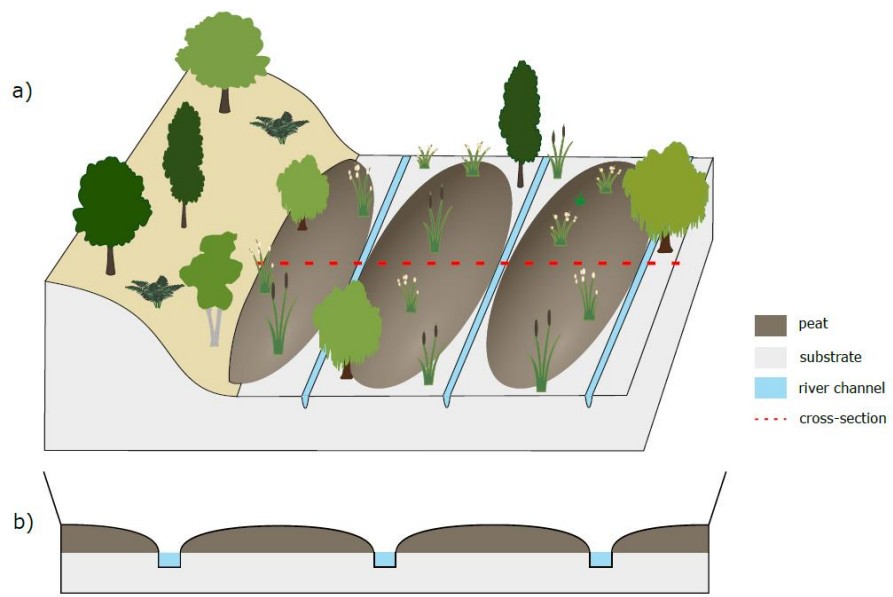

**Figure 3: Conceptual drawing of the model domain. a) Landscape setting of the alluvial peatland, represented by elliptic bogs on top of an impermeable substrate with straight river channels. The red dotted line indicates the cross-sectional location. b) Schematic floodplain cross-sectional drawing. The 1D model simulates the peat profile at the centre of a single elliptic bog.**





### 3.1 Local peatland model

The alluvial peat growth is modelled using a modified version of the Digibog model (Morris et al., 2011). To calculate the
water table dynamics, Childs' equation for an elliptic bog is used with the length of the ellipse (parallel to the direction of the
river channel flow) being infinitely long relative to the width (fig. 3) (Childs, 1969). Additionally, changes were made to the
calculation of the potential and actual evapotranspiration rates because of the important differences in vegetation type between
raised bogs and alluvial peatlands. The annual potential evapotranspiration rate is calculated using the Thornthwaite equation
based on mean monthly temperatures because of its relative simplicity and the low amount of input data needed (Moeletsi et
al., 2013). The potential evapotranspiration rate is subdivided in soil evaporation and plant transpiration rates based on the leaf
area index (Williams et al., 1983). The actual evapotranspiration rates are calculated based on the water table depth, plant
rooting depth and Ellenberg indicator value for moisture (f-value). As such, the local vegetation characteristics are taken into
account (see appendix, section A2 for details).

The biomass productivity equation used in the original Digibog model is constructed for typical bog-building species such as
*Sphagnum* mosses. Here, the net primary productivity is calculated using the Thornthwaite Memorial equation, which does
not assume the presence of a specific vegetation type and has been successfully applied in other peatland models (Heinemeyer
et al., 2010). The Thornthwaite Memorial equation has been constructed for global applications and thus allows for a wider
variety of vegetation types (Lieth and Box, 1972). Since the actual evapotranspiration rate is dependent on both the water table
depth and vegetation characteristics, the peatland vegetation indirectly also influences the calculated biomass productivity.

### 3.2 River basin hydrology model

The river basin hydrology is modelled using the STREAM model, which is a grid-based, spatially distributed water balance
model (Aerts et al., 1999). In STREAM, the hydrological cycle of a basin is simulated on a raster, where each grid cell consists
of a set of fluxes and reservoirs. Water is added to the system by precipitation. Rainfall can either contribute directly to the
discharge as surface runoff or is added to the soil reservoir. When temperatures at a certain location are below 0°C, the
precipitation is added to the snow reservoir. This reservoir can contribute to the precipitation through snowmelt using a degree
day factor model. When the precipitation reaches the soil surface, the amount of generated runoff is calculated using the curve
number approach, making the runoff amount dependent on the soil type and land cover. The water that enters the soil is added
to the soil reservoir as long as the soil water content is below the field capacity. Otherwise, it flows to the deep groundwater
reservoir. The contribution of the soil and groundwater reservoirs to the total discharge are dependent on the amount of water
stored in the reservoir at each location, a calibration parameter and the local slope (Aerts and Bouwer, 2003). Each gridpoint
can thus contribute water to the basin discharge using three different pathways (surface runoff, soil throughflow and
groundwater flow). The total discharge of all grid points is accumulated using a flow accumulation algorithm. As such, the
discharge is not routed explicitly through the landscape, but is assumed to accumulate according to the flow network at the





surface and to leave the river basin at the outlet (Aerts and Bouwer, 2003) (details on the use of STREAM and the calibration
procedure are given in the appendix, section A1).

### 3.3 Coupling the STREAM and modified Digibog models

The daily discharge time series simulated by STREAM are used as a lateral boundary condition for the adapted Digibog model. The total discharge is divided equally over all river channels and converted to water levels using Manning's equation. The position of the channel(s) is assumed to be fixed. As a consequence, the lateral extent of the peatland is assumed to remain constant over the entire simulation period. When simulating the peatland development, the discharge-water level relationship changes over time for discharges above bankfull stage due to changes in the peat surface elevation. As a result, this relationship is updated every time step. To apply Manning's equation, roughness values of 0.035 for the channels and 0.068 for the floodplain surface were used (Hosia, 1980; Lappalainen et al., 2010; Marttila et al., 2012; Medeiros et al., 2012a; Thomas and Nisbet, 2007; Tuukkanen et al., 2012). The floodplain slope is determined by calculating the mean floodplain gradient over a distance of 1,000 metres up- and downstream of the studied location using LIDAR elevation data with a 1-metre resolution. The assumption is made that the floodplain slope at a specific location did not change significantly throughout the Holocene. Since there is no significant relationship between Holocene floodplain stratigraphy thickness and catchment area for both the Dijle and Grote Nete rivers, the current floodplain slope can be assumed to be representative for the entire Holocene. The local peat growth model simulates peatland development at millennial timescales, but due to computational limitations, this was not possible for the STREAM model. To overcome this issue, the river basin hydrology was simulated for a 100-year period and the model output is repeated every 100 years until the simulation time of the local peat growth model is met.

### 4 Model applications

#### 4.1 Sensitivity analysis

First of all, an OAT (one-at-the-time) sensitivity analysis was performed for the modified Digibog model by varying the different model parameters over a specified range, which is determined by a review of the literature. Only the parameter under consideration is varied stepwise over 75% of the range mentioned in the literature, while all others are kept at their standard value (a detailed table with the simulated range for each of the parameters is listed in the appendix, section A4). The local peatland model is run under conditions typical for the alluvial peatlands in northern Belgium for a time period of 10,000 years, which is sufficiently long to reach a peat thickness in equilibrium with the simulated conditions. The sensitivity was analysed based on the peat thickness at the end of the simulation period. Parameter values for the calculation of the potential and actual evapotranspiration rates are derived from literature. Due to the limited values available, these parameters are not included in the sensitivity analysis. The model domain of the 1D Digibog model only includes the peat layer, with boundary conditions set by the impermeable substrate and the water level in the adjacent channels. However, in an alluvial setting, the peat hydrology can be influenced by external factors such as channel network geometry and river basin hydrology. To test the



sensitivity of the model to external hydrological factors, two additional parameters were varied. Firstly, the substrate below the peat was no longer assumed to be impermeable, but an additional vertical water flux was incorporated, representing the effect of a hydrological interaction between the peat and its substrate in both upward and downward direction. This flux is incorporated as an additional source or sink term, similar to the precipitation. Secondly, the effect of changes in the lateral extent of the peat layer was studied. The lateral extent specifies the distance between the channel and the centre of the peat

body and is thus determined by the number of channels in a floodplain cross-section and their spacing.

The results of the OAT sensitivity analysis indicate a moderate impact of peat properties on the final peat thickness after 10,000 years of simulation, ranging from 1.46 to 3.56 metres (fig. 4a). Variations in the oxic decomposition rate result in a decrease in the final peat thickness for increasing rates, although the effect is relatively limited in comparison to other variables. The anoxic decomposition rate does not strongly influence the thickness, except for very low values of anoxic

decomposition, which lead to lower peat thickness values. This can probably be attributed to the relationship between the degree of decomposition and the hydraulic conductivity of the peat, where low anoxic decomposition rates lead to higher conductivity values, increasing the exposure of the peat column to oxic conditions. With respect to the physical properties, the peat thickness is mostly influenced by the dry bulk density and the a-parameter of the hydraulic conductivity relationship (especially in the lower part of the simulated range). The a-parameter determines the conductivity for highly decomposed

peat. Since most of the peat profile consists of well decomposed peat, this parameter determines the capacity of the peat column to keep itself water saturated. As a result, low values for the a-parameter lead to higher final thickness values.

The climatic variables, on the other hand, have a stronger effect on the simulated peat thickness, with a strong positive relationship with the mean annual precipitation and a slight negative relationship with the mean annual temperature (fig. 4b). This indicates that the increased biomass productivity due to higher temperatures does not compensate the temperature effects

on the evapotranspiration and peat decomposition.



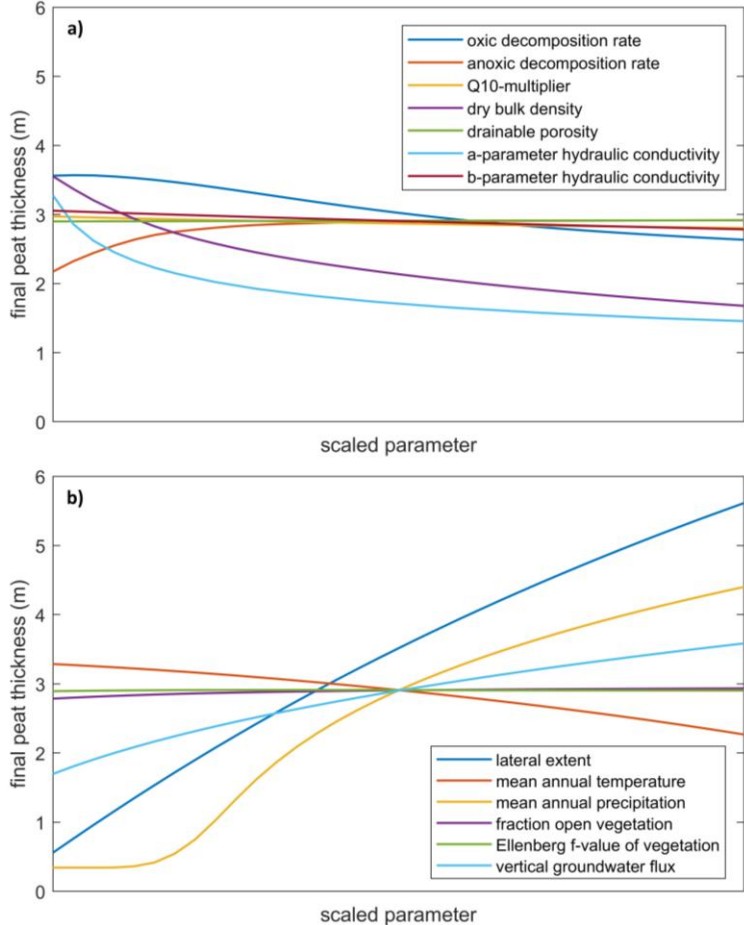

**Figure 4: Final peat thickness (m) for (a) all variables used in the one-at-a-time (OAT) parameter sensitivity test for model parameters related to peat properties; and (b) external factors including the peatland vegetation, climatic conditions and hydrology. Each parameter is varied over the range mentioned in table A6 in the appendix.**

In contrast to the climatic conditions, the parameters related to the local floodplain vegetation (fraction open vegetation and

Ellenberg f-value) have a much smaller effect on the peat thickness. Overall, a tree-rich vegetation type appears to have a

slight positive effect on the final peat thickness in comparison to an open vegetation type, suggesting that the increased

biomass productivity of trees more than compensates the increase in plant transpiration. The two model parameters related to

the hydrological setting of the alluvial peatland (lateral extent and vertical groundwater flux) result generally in the largest

differences in the final peat thickness. An upward vertical flux from the substrate to the peat layer results in an increased peat

thickness due to the increased amount of water feeding the peat layer. The lateral extent is positively correlated with the final

thickness. A larger distance between the channels lowers the slope of the groundwater mound in the peat layer, reducing the

drainage efficiency.





## 4.2 Local peatland model

A first model application is focusing on local alluvial peatland dynamics, i.e. without taking into account the effect of the river basin hydrology and river channel dynamics. This local peatland model is applied to the locations of Korbeek-Dijle and Hulshout and run from 10.05 ka BCE until now, assuming a fixed water level in the channels at the level of the substrate, with a channel spacing of 200 metres. The vegetation on top of the peat profile is assumed to be a mixture of trees and open vegetation, each accounting for 50 percent of the areal cover. Time series for temperature and precipitation, were constructed

using a pollen-based climate reconstruction with a spatial resolution of 1° x 1° and a temporal resolution of 500 years, expressed as anomalies relative to the year 1850 CE (Mauri et al., 2015). Annual time series were constructed by selecting randomly daily time series with a length of one year from the 30-year climatic period around the year 1850 CE for the station of Ukkel (Belgium). The time series were corrected in such a way that the mean value matches the pollen-based climate reconstructions. Random variability was added to the mean annual value, equal to the observed standard deviation for the period 1835 – 1864

in Ukkel. For the mean annual precipitation amount and mean annual temperature, the relative and absolute standard deviation were used, respectively.

The simulated peat thickness evolution shows a phase of rapid peat growth up to 2 to 2.5 metres between 10.05 and 6.05 ka BCE after which a much lower growth rate leads to final peat thicknesses of approximately three metres by the Middle Ages (fig. 5). The loess belt (Dijle) and sand belt (Grote Nete) show a similar peatland development trajectory, with some minor

differences, which can be attributed to slightly different Holocene climate reconstructions for both regions. The similarity between both trajectories, however, does not match the observed differences in floodplain stratigraphy as derived from field data (Swinnen et al., 2020). The floodplain stratigraphy of the Dijle river shows a clear transition from alluvial peatlands to mineral overbank sedimentation with a mean compaction-corrected peat thickness of 1.6 metres. In contrast, the Grote Nete floodplain stratigraphy is highly variable with alternating layers of peat, organic-rich sediments and mineral sediment and a

lower mean peat thickness of 0.56 metres.



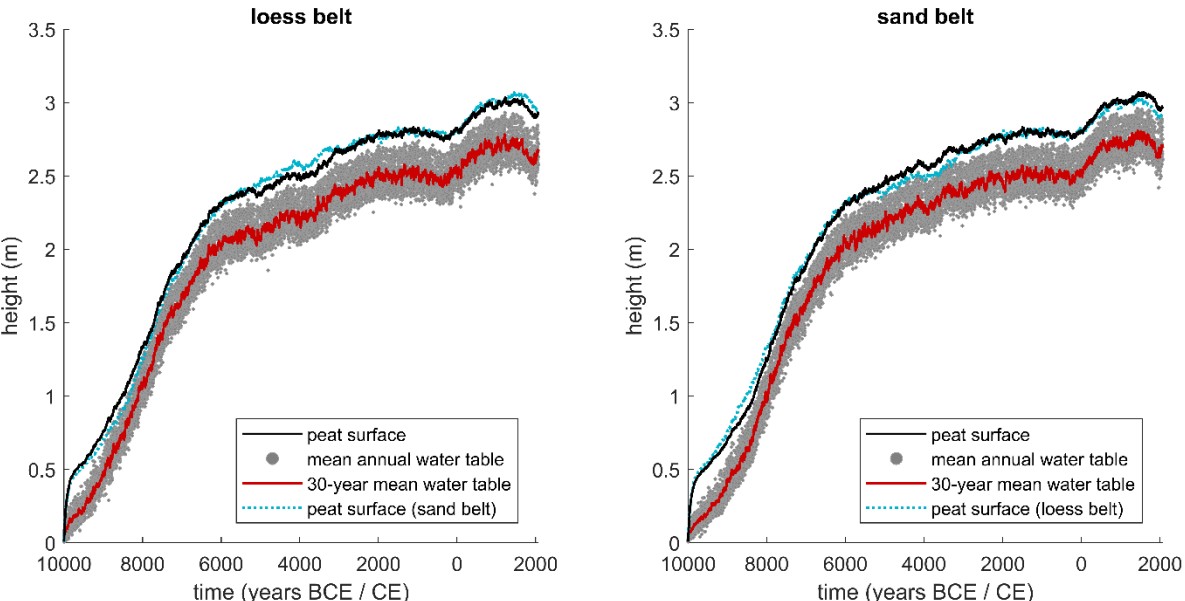

**Figure 5: Simulated evolution of an alluvial peatland for both the loess and sand belts since 10.05 ka BCE.**

**4.3 Environmental conditions and river network characteristics**

In a next step, the calibrated STREAM model was applied to the Dijle and Grote Nete river basins taking into account changes
in river basin hydrology following climate and land use change. The climate scenarios are constructed in such a way that they
cover the entire range of temperature-precipitation combinations, as were present during the period 12 ka – 100 cal BP. Five
points were selected along the climate trajectory of both river basins as derived from the pollen-based climate reconstruction
(climate scenarios 1 – 5) (Mauri et al., 2015). In addition, a sixth scenario was added, representing the average mean annual
temperature and mean annual precipitation combination for the period 12 ka – 100 cal BP (fig. 6). High-resolution time-series
were constructed using the same approach as for the application of the local peatland model.





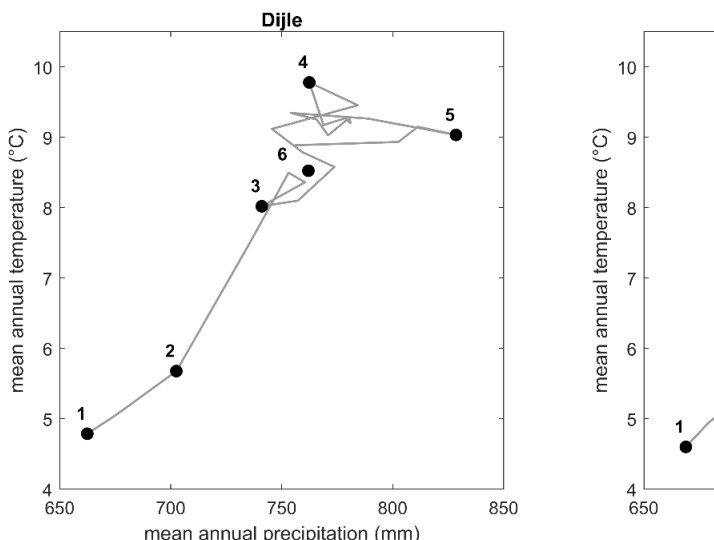
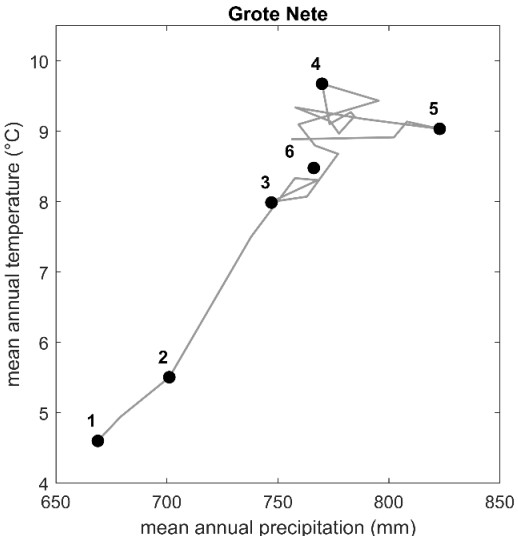

**Figure 6: Mean annual precipitation (mm) and mean annual temperature (°C) for each of the six climate scenarios for both the Dijle and Grote Nete rivers. The grey line indicates the climate evolution (12 ka – 100 cal BP) according to the pollen-based climate reconstruction (Mauri et al., 2015).**

To represent the land cover, five scenarios were constructed which consist of three land cover types (forest, grassland/short vegetation and cropland/bare soil) with varying cover fractions. The scenarios range from a fully forested landscape to an open landscape dominated by cropland (table 1). As the scenarios are designed to cover the period 12 ka – 100 cal BP and widespread man-made structures are a rather recent phenomenon, built-up area is not included in the scenarios. Details on the allocation of the land cover types is described in the appendix, section A1.2. This results in 30 possible climate – land cover scenario combinations. For each of these, STREAM was run for the Dijle and Grote Nete basins for a period of 100 years.

**Table 1: Vegetation fractions (%) for each of the five land cover scenarios.**

| Land cover scenario | Forest | Grassland/short vegetation | Cropland/bare soil |
|:---:|:---:|:---:|:---:|
| Scenario 1 | 100% | 0% | 0% |
| Scenario 2 | 66.6% | 16.7% | 16.7% |
| Scenario 3 | 33.3% | 33.3% | 33.3% |
| Scenario 4 | 0% | 50% | 50% |
| Scenario 5 | 0% | 33.3% | 66.7% |

Whilst much research has been done on changing Holocene alluvial stratigraphies, the available information on river planform, river geometry and on the position of the channel relative to the alluvial peat layer is limited for alluvial peatlands (Broothaerts et al., 2013; Candel et al., 2017; Lespez et al., 2015a; Nanson, 2009). As a result, it is not possible to identify a single spatial configuration for the model domain which can be assumed to be representative. Therefore, the coupled model is run to test the influence of channel morphology on alluvial peat growth. Firstly, different scenarios were constructed whereby the number of





channels ranges between 1 and 25, thus simulating planforms ranging from single-channel meandering to multi-channel anastomosing or anabranching wetland systems. Next, the scenarios for the channel dimensions were made dependent on the

number of channels. This was done because it is unrealistic that a single-channel river has a very small cross-sectional area or that in a floodplain with a large number of channels, all will have a large cross-sectional area. As a result, for each number of channels, five possible channel dimensions were constructed, with an overall decrease in the cross-sectional area of each channel with an increasing number of channels (fig. 7). All floodplain channels are assumed to be rectangular, with a specified width and depth. A study by Nanson et al. (2010) found a mean width/depth-ratio of 2.2 for a set of alluvial peatland channels,

approximating the value of 2, which is the ratio for the most efficient water transport in rectangular channels without bedload. As a result, the dimensions scenarios assume a width/depth-ratio of two for three out of the five scenarios, with varying cross-sectional area (small, medium and large cross-sectional area). To test the effect of the width/depth-ratio on the resultant peatland development, two additional scenarios are constructed with width/depth-ratios of one and four with the same cross-sectional area as the middle scenario out of the three scenarios with a width/depth-ratio of two (fig. 7).

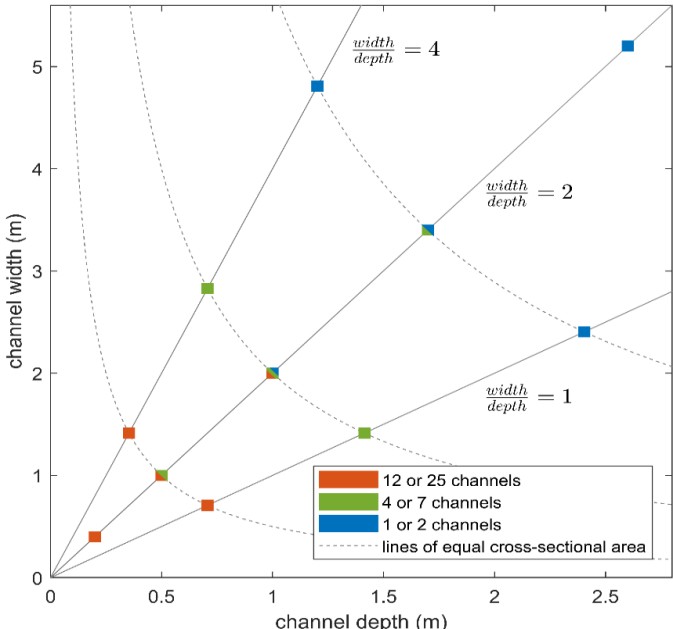

**Figure 7: Channel width (m) and depth (m) for the different channel dimensions-scenarios. For each value for the number of channels, five possible channel dimensions scenarios are used.**

Finally, not only a setting with the river incised in the substrate was tested, which is the standard situation for the climate and land use runs, but also a setting whereby the river channel is located in the peat layer itself, with the channel bottom at the

level of the base of the peat column. The same scenario combinations as before were run for these two contrasting settings, with the difference that the channel dimension scenarios for the second setting only take into account the width of the channel (fig. 8).





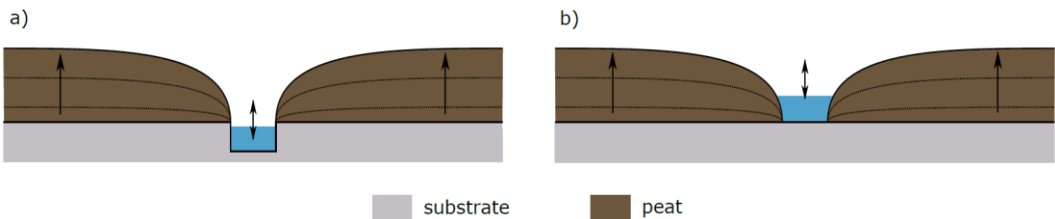

**Figure 8: Sketch of the two conceptual channel models used for the long-term peatland development simulations. a) Rectangular**
**channel, situated within the substrate, with the top of the channel corresponding to the base of the peat. b) The channel located on**
**top of the substrate, with the channel bottom corresponding to the base of the peat. The channel walls consist of the sloping sides of**
**the peat bodies.**

Over all scenario combinations, the simulated peat thickness ranges between 0.77 and 9.52 metres, with a mean value of 3.62

metres for the Dijle river and between 0.89 and 10.73 metres, with a mean value of 4.20 metres for the Grote Nete river. The

results indicate that especially the number of river channels strongly influences the peatland development, with a strong

decrease in peat thickness with an increasing number of channels (fig. 9).

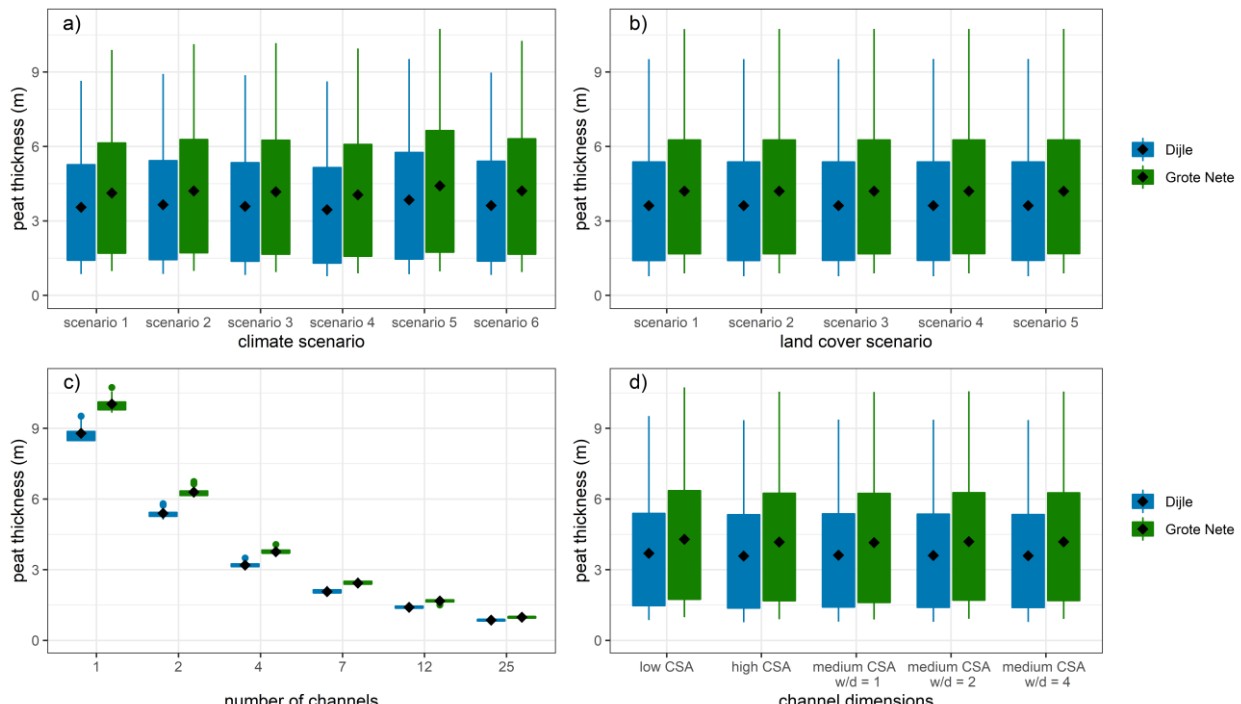

**Figure 9: Boxplots of the simulated peat thickness (m) for all scenario combinations over a time period of 10,000 years for the Dijle**
**and Grote Nete rivers, subdivided per climate scenario (a), land cover scenario (b), number of channels (c) and channel dimensions**
**(d). The mean value is indicated by a black diamond. The box indicates all values within the 25th to 75th percentile range and the**
**whiskers represent all values within the range from the 25th percentile – 1.5\*interquartile range to the 75th percentile +**
**1.5\*interquartile range. Coloured dots represent all other values outside this range.**

When only considering all scenarios with four river channels, the effect of climate, land cover and channel dimensions on the

peat thickness can be evaluated (fig. 10). The peat thickness shows to be the highest under climate scenario 5, which has the





highest mean annual precipitation amount, and lowest under climate scenario 4, which has the highest mean annual temperature (fig. 10a). The different land cover scenarios do not result in significantly different peat thickness values (fig. 10b). The dimensions of the river channels have a minor effect on the simulated thickness, with the highest values for channels with a small cross-sectional area and lower thickness values for channels with a larger cross-sectional area. The three dimension-scenarios with a medium cross-sectional area but with varying width/depth-ratios shows a small negative effect of increasing width/depth-ratios on the resultant thickness (fig. 10c).


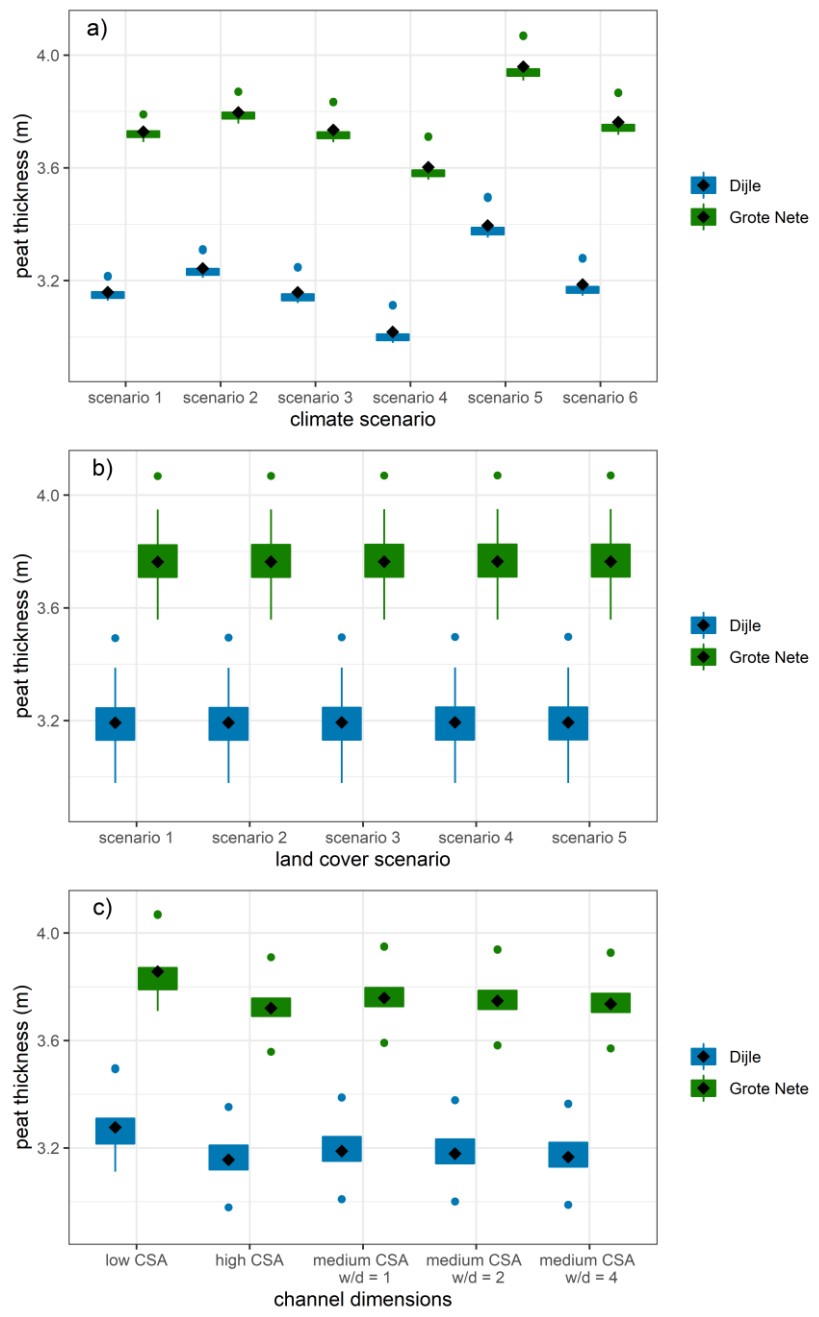

**Figure 10: Boxplots of the simulated peat thickness (m) for all scenario combinations with four river channels, over a time period of 10,000 years for the Dijle and Grote Nete rivers, subdivided per climate scenario (a), land cover scenario (b) and channel dimensions (c). The mean value is indicated by a black diamond. The box indicates all values within the 25th to 75th percentile range and the whiskers represent all values within the range from the 25th percentile – 1.5\*interquartile range to the 75th percentile + 1.5\*interquartile range. Coloured dots represent all other values outside this range.**






Additionally, when comparing the two conceptual channel configuration scenarios, an overall increase in simulated peat thickness is observed for the simulations where the channels are located on top of the substrate, relative to those where the

channel is situated in the substrate. The mean simulated peat thickness over all scenarios increases from 3.62 to 3.72 metres for the Dijle river and from 4.20 to 4.30 metres for the Grote Nete river.

### 4.4 River channel properties

Whilst the STREAM model determines the upstream hydrological boundary conditions in the alluvial setting, local hydrology is also determined by channel and floodplain properties such as the roughness and slope. For instance, lower roughness values

and higher slopes will lead to a more efficient drainage of the alluvial wetlands, potentially leading to lower peat growth rates. In total, six roughness scenarios were constructed where both channel and floodplain roughness are varied together over 75% of the range found in the literature (table 2). The floodplain slope is varied over 75% of the range in slopes observed across the Dijle and Grote Nete floodplains in five scenarios. The middle-of-the-road climate and land cover scenarios were used, which corresponds to climate scenario 6 and land cover scenario 3.

**Table 2: Range over which the channel and floodplain roughness (s m$^{-1/3}$) and the channel/floodplain slope (m m$^{-1}$) are varied in the scenario analysis.**

| Parameter | Minimum value | Maximum value | References |
|---|---|---|---|
| Channel roughness | 0.02 | 0.07 | (Hosia, 1980; Lappalainen et al., 2010; Marttila et al., 2012; Tuukkanen et al., 2012) |
| Floodplain roughness | 0.035 | 0.15 | (Medeiros et al., 2012b; Thomas and Nisbet, 2007) |
| Channel/floodplain slope | $2.5*10^{-4}$ | $2.5*10^{-3}$ | / |

The simulation results indicate a limited but positive effect of increasing channel and floodplain roughness on the resultant peat thickness (fig. 11). Although the floodplain slope is varied over an order of magnitude, also here, the effect on the

simulated peat thickness is rather limited.



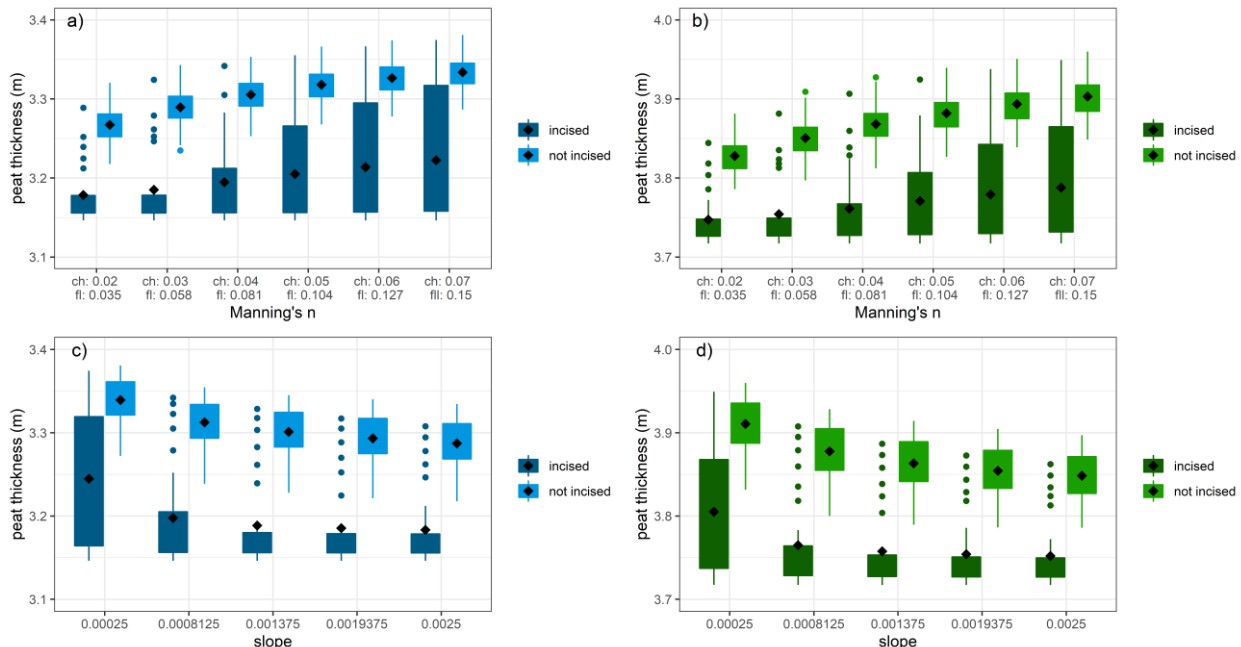

**Figure 11: Boxplots of the simulated peat thickness (m) after 10,000 years of simulation for the Dijle (a, c) and Grote Nete (b, d) rivers for the different scenarios of channel (ch) and floodplain (fl) roughness coefficients (a, b) and the floodplain and channel slope (c, d) for a floodplain setting with four channels and for the five possible channel dimension-scenarios. Each plot divides the results between the scenarios with a rectangular channel, incised in the substrate and the scenarios without an incised channel. The mean value is indicated by a black diamond. The box indicates all values within the 25th to 75th percentile range and the whiskers represent all values within the range from the 25th percentile – 1.5\*interquartile range to the 75th percentile + 1.5\*interquartile range. Coloured dots represent all other values outside this range.**

**4.5 Vertical channel aggradation**

In all previous model runs, the position of the river channel(s) was fixed in space, both in lateral and vertical direction. While the detailed Holocene history of the channels in these river basins is not yet clear and requires specific reconstructions, the effect of vertically aggrading channels was simulated by increasing the channel bottom elevation at a fixed rate. Here, the vertical aggradation rate of the channel was set to the mean Holocene peat accumulation rate, calculated as the mean Holocene peat thickness over the entire river basin divided by the duration of the Holocene (since 11.7 ka cal BP). This results in an aggradation rate of 0.17 millimetres per year for the Dijle river and 0.05 millimetres per year for the Grote Nete river.

The results indicate a beneficial effect of vertically aggrading channels on the resultant peat thickness, with higher simulated thickness values for both the Grote Nete and Dijle river (fig. 12). The effect is more pronounced for the Dijle river due to the higher aggradation rate, and affects mostly the scenarios with a high number of channels. For the Dijle river in a setting with one river channel, the mean simulated peat thickness increases with 1.17 metres from 8.79 to 9.96 metres. For a setting with 25 channels, the mean thickness increases with 1.66 metres from 0.86 to 2.51 metres.





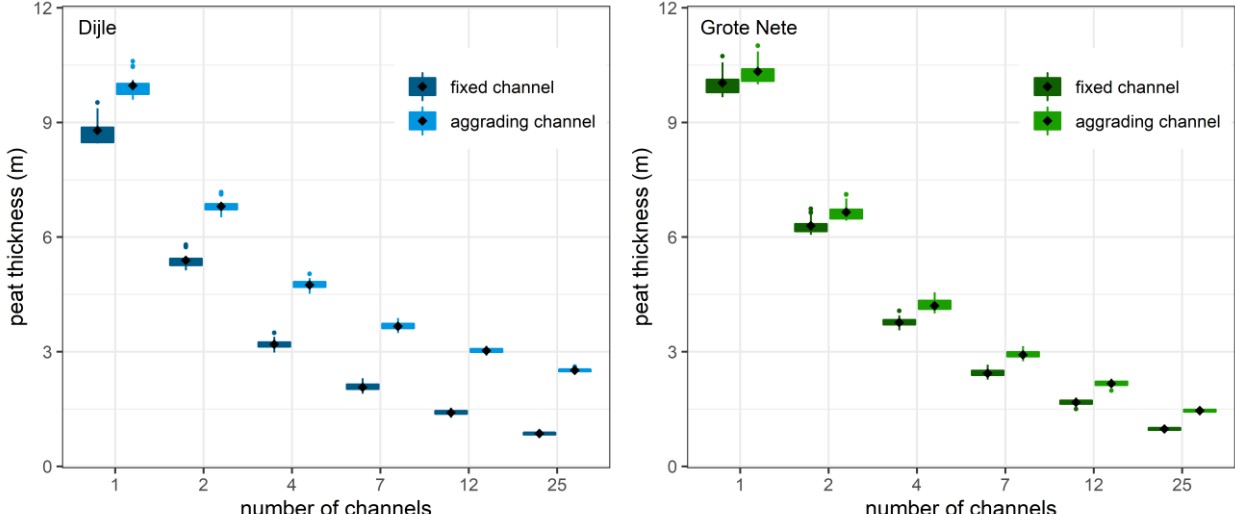

**Figure 12: Boxplots of the simulated peat thickness (m) over all scenario combinations after 10,000 years of simulation for the Dijle and Grote Nete rivers for different numbers of channels. The simulations assume channels with a fixed position in the substrate (*fixed channel*) or channels which aggrade vertically at a fixed rate (*aggrading channel*). The mean value is indicated by a black diamond. The box indicates all values within the 25th to 75th percentile range and the whiskers represent all values within the range from the 25th percentile – 1.5\*interquartile range to the 75th percentile + 1.5\*interquartile range. Coloured dots represent all other values outside this range.**

## 5 Discussion

A new modelling framework was presented to simulate the development of alluvial peatlands over Holocene timescales by coupling a local peat growth model with a river basin hydrology model. The 1D Digibog model was modified to better represent peat growth in floodplain environments. In a first step, the modified Digibog model was used to simulate the Holocene development of the alluvial peatlands for the two contrasting study areas, excluding the effect of the river basin hydrology. The results show a rapid peatland development in accordance with increasing temperatures and precipitation amounts during the early Holocene, stabilising from approximately 6000 BCE onwards. The simulated development trajectories match relatively well with the field data in terms of the timing of peatland development. The peat growth between 10.05 and 6.05 ka BCE in the Belgian river valleys corresponds with reconstructed Surface Peat Index curves, which indicate a strong increase in the floodplain area under active peat growth until 8 ka cal BP, especially for the rivers in the loess belt (Swinnen et al., 2020). However, the simulations show a rather similar Holocene trajectory for the Dijle and Grote Nete rivers, with only minor differences in terms of peat thickness and timing of changes, which does not correspond with the available data on floodplain stratigraphy (fig. 5). The high degree of similarity between both areas can be attributed to the modelling assumptions. As only the local peat growth model is used, the potential differences due to processes at the basin scale or in the local river network cannot be simulated. As such, these results show that a model setup which only simulates the internal dynamics of alluvial





peatlands and that does not incorporate external factors such as seepage, river channel dynamics and river basin hydrology seems to be unable to reconstruct the observed differences in alluvial peatland dynamics between the loess and sand belts.

## 5.1 Factors controlling alluvial peatland dynamics

Sensitivity analysis was used to determine the effect of individual model parameters on the resultant peatland dynamics to identify important processes controlling the long-term dynamics of alluvial peatlands. One of the important changes to the original Digibog model is the incorporation of wider variety of vegetation types. Most long-term peatland models use empirical equations which relate the productivity to one or more hydrological or vegetation parameters such as the actual evapotranspiration rate or the water table depth. However, the limited data availability at Holocene timescales and corresponding simplicity of these equations results in a trial-and-error procedure of selecting the most appropriate productivity equation. As such, the productivity equation used here has been applied across the globe and allows a wide variety of vegetation types (Lieth, 1973; Lieth and Box, 1972). While this approach might be less precise than more detailed equations, the results of the sensitivity analysis indicate that the model parameters related to the peatland vegetation have a limited effect on the resultant peat thickness. Contrary to local floodplain vegetation, physical properties of peat including the dry bulk density and the hydraulic conductivity, do show to have a strong impact on final peat thickness (fig. 4). In addition, several model parameters related to the local hydrological setting such as precipitation, spacing between the channels and vertical groundwater flux also show to be very influential. This demonstrates the importance of detailed environmental reconstructions of past conditions and the need for a correct representation of the hydrological interaction between the alluvial peat layer and its surroundings. Although a detailed calibration and validation procedure was not possible for the peat growth model, the results of the sensitivity analysis can be compared to compaction-corrected peat thickness values for both river basins, derived from a dataset of soil coring data (Swinnen et al., 2020). The range in peat thickness values, simulated in the sensitivity analysis (0.4 – 5.61 metres) matches more or less with the reconstructed uncompacted peat thickness data (0.1 – 6.7 metres) for the different river valleys. The results of the scenario-based simulations demonstrate that changes in the river discharge, related to climatic and land cover changes, have a limited effect on the alluvial peatland development (fig. 9 and fig. 10). This can be attributed to the fact that climate and land cover changes mostly affect the magnitude of peak events, rather than the mean discharge. As peak flows are relatively rare by nature, their effect on the peatland water table and thus on the thickness of the peat layer is rather limited. In addition to the hydrological effect, changes in climate or land cover over the upstream basin will also affect other aspects such as landscape sediment dynamics, river channel stability and groundwater-peatland interactions, which are not simulated here.

### River channel dynamics

Overall, the results of the scenario analysis indicate that the characteristics related to the position of river channel(s) are the most influential, especially the number of channels in a floodplain cross-section (fig. 9 and fig. 12). A higher number of channels in the same floodplain cross-section increases the slope of the water table between the centre of the peat body and





the river channel, resulting in an increased drainage efficiency and lower simulated peat thickness values. Both the OAT analysis and scenario-based simulations demonstrate the important effect of model parameters related to the local river network properties on the simulated peat thickness. Especially for settings with multiple river channels the simulated peat thickness is more in line with the mean measured peat thickness of 1.60 metres for the Dijle river and 0.56 metres for the Grote Nete river. These results suggest that the early Holocene river network in both the Grote Nete and Dijle river basins did not consist of a

single-channel situation as is the case in current time periods but rather an anastomosing pattern with multiple active channels and peat growth on the islands in between. The formation of an anastomosing pattern can be explained by the low floodplain gradients in both the Dijle and Grote Nete river basins and the erosion resistance of peat layers. In such a river system, the formation of new channels is triggered by avulsions, which can be caused by obstructions such as log jams or beaver dams (Diefenderfer and Montgomery, 2009; Gradziński et al., 2003; Makaske, 2001; Polvi and Wohl, 2012; Stefan and Klein, 2004).

The setting of alluvial peatlands with an anastomosing river network have been described in other studies across the European lowlands (Broothaerts et al., 2014a; Gradziński et al., 2003; Lespez et al., 2015b). On the other hand, Candel et al. describe a different floodplain development trajectory for a peat-filled lowland stream (Drentsche Aa) where the difference in erosion resistance of the peat and the valley sides result in oblique aggradation and highly sinuous single-channel planforms (Candel et al., 2017). However, floodplains of the Drentsche Aa contain peat layers of up to seven metres thick, which is much more

than the thicknesses measured for the Dijle and Grote Nete rivers. This combination of a single-channel setting and high peat thickness values for the Drentsche Aa corresponds with the model simulations in this study assuming a limited number of channels and vertical channel aggradation (fig. 12). These results seem to suggest that the overall peat thickness can provide a rough estimate of the typical floodplain drainage pattern. However, which factors determine the presence of a single-channel or anastomosing pattern in peat-filled valleys is unclear. The amount of available studies is rather limited and does not allow

to identify which setting was more common in the European lowlands throughout the Holocene.

The two conceptual channel configurations, which are applied here, determine the effect of the channel hydrograph on the peatland water table (fig. 11). When considering an incised channel, only discharge events above bankfull discharge will influence the drainage in the peat layer, given the assumption of an impermeable substrate below the peat. If the channel is situated on top of the substrate, all water level variations inside the channel will influence the water level in the peat layer.

Overall, the results indicate a modest increase in the simulated peat thickness for the configuration where the channel is located on top of the substrate, relative to an incised channel. In addition, other parameters representing channel properties such as the channel roughness and slope have a limited effect on the simulated peat thickness compared to other model parameters. Vertical aggradation on the other hand has a much more profound impact on the resulting peat thickness, especially for the scenarios with multiple channels, suggesting that channel mobility rather than channel properties impact peat growth (fig. 12). However,

these scenarios assume a vertically aggrading channel without geomorphic interaction between the peatland and the river channel. While peat is rather cohesive and thus limits channel mobility, it is unclear how realistic this assumption is over long timescales. Overall, the model simulations suggest that alluvial peatland development is strongly determined by the spatial organisation of the river channels across the river floodplain and the vertical channel mobility over longer timescales. As such,





a good understanding of the geomorphic and hydrologic interactions between the river network and the alluvial peatlands is
required to make detailed simulations of past and future alluvial peatland dynamics.

## 6 Conclusion

In this study, a new model was presented, specifically designed to simulate alluvial peatland development over Holocene
timescales in relation to changes in both local and regional environmental conditions. A scenario-based approach was used to
assess the sensitivity of alluvial peat growth to environmental changes under a wide range of settings. Although the simulations
are explorative, the results demonstrate that the approach used here can improve our understanding of the different interactions
and feedbacks between alluvial peatlands and the river network. Overall, the alluvial peatland dynamics appear to be strongly
determined by the setting and dynamics of the local river network, rather than by internal peatland dynamics or regional
environmental changes. The scenario analysis suggests that a floodplain setting with an anastomosing river pattern and peat
formation on the islands in between the channels matches best with the observed peat thickness in the studied river basins in
the European loess and sand belts. In general, these results highlight the need for detailed reconstructions of Holocene
floodplain and river channel dynamics, which are required for a detailed understanding of past and future peatland dynamics
in alluvial environments.





## Appendix

**A1 STREAM water balance model**

The STREAM model is grid-based water balance model which simulates the hydrological cycle of a river basin using a set of spatially explicit reservoirs and fluxes (fig. A1). A detailed discussion of the model features is given by Ward et al., 2007. Here, the modifications to the original model layout, the input data and the calibration procedure are discussed.

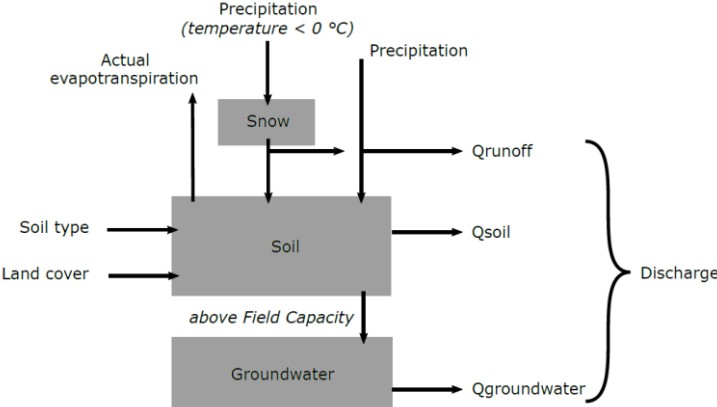

**Figure A1: Schematic structure of the STREAM water balance model (based on Aerts et al., 1999).**

While STREAM is a spatial model, the produced runoff is not explicitly routed through the landscape, but is assumed to accumulate according to the steepest descent principle and reach the river basin outlet during the same time step. Given the relatively short timestep of one day used in this study, this assumption results in very high peak discharge values during rainfall events. Here, the travel time between each grid cell and the basin outlet was calculated by defining the downstream distance

to the nearest stream and the in-stream distance to the outlet of the basin. By selecting specific values for the velocity of both overland runoff (0.3 metres per second) and channel flow (0.45 metres per second), the time necessary for runoff to reach the basin outlet could be calculated for each location. The assumption was made that all precipitation falls in the middle of each time step. As a result, for each grid cell, it can be calculated during which time step the overland flow reaches the outlet, based on the travel time. This results in an attenuation of the peak discharge, with runoff generated further upstream in the basin

being added to the basin hydrograph in later time steps.

**A1.1 STREAM model calibration**

In total, the STREAM model contains three calibration parameters which allow to fine-tune the model behaviour. The evapotranspiration parameter ($P_{cal}$) calibrates the amount of evapotranspiration and thus controls the balance between the basin precipitation and the river flow at the basin outlet. The other two calibration parameters ($S_{cal}$ and $G_{cal}$) determine the

relationship between the amount of water present in the soil and groundwater reservoirs and their contributions to the basin





discharge. The model is calibrated for both the Dijle and Grote Nete basins at a spatial resolution of 50 metres, by comparing the simulated daily discharge time series with the observed discharge at the location of available gauging stations. In this case, the gauging stations of the Flemish Environmental Agency (VMM) in Sint-Joris-Weert (Dijle) and Hulshout (Grote Nete) are used (fig. A2, table A1). Simulations are performed for the period for which observed discharge time series are available,

extended with a two-year spin-up phase to allow the different water reservoirs in the STREAM model to fill up. All three calibration parameters are varied stepwise and the evaluation of the best fitting parameter combination is based on the Nash-Sutcliffe model efficiency (Nash and Sutcliffe, 1970).

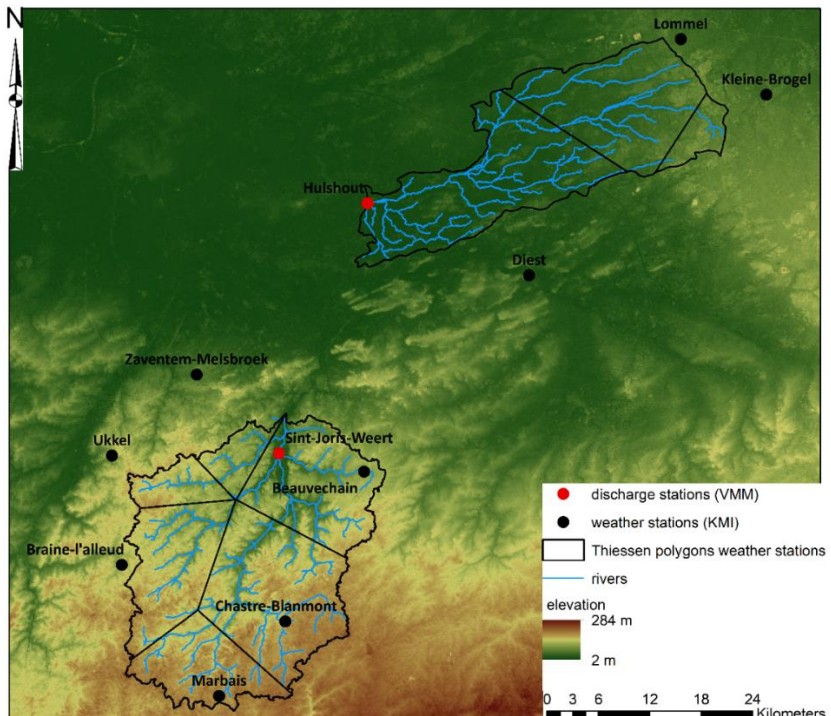

**Figure A2: Map of the Dijle and Grote Nete basins, with indications of the discharge measuring stations (VMM) and weather stations**
**(KMI) used for the calibration of the STREAM model. The river basins are subdivided by Thiessen polygons, based on the location of the weather stations.**

In the model simulations, it is assumed that all discharge produced by the river basin passes through the channel at the outlet of the basin. This assumption can be made for the observed time series as well, except when the measured discharge exceeds the bankfull stage, which is the upper limit for the observed discharge time series. As a result, the calibration process excludes

the time steps in which the modelled values exceed the bankfull discharge value at the gauging station location, since the observed value will never exceed this threshold by nature. The calibration simulations span the period 1977 – 2017 for the Dijle river and 1985 – 2017 for the Grote Nete river. The climate time series (mean daily temperature and daily precipitation) are derived from weather stations of the Belgian Royal Meteorological Institute (KMI) for stations in and around the river basins. These point data are converted to rainfall and temperature maps using Thiessen polygons (fig. A2 and table A1).





**Table A1: List of the hydrological and meteorological stations used for the calibration of the STREAM model for the Dijle and Grote Nete river basins.**

| River basin | Calibration period | Hydrological station | Meteorological stations used for precipitation data | Meteorological stations used for temperature data |
|---|---|---|---|---|
| Dijle | 1973 - 2017 | Sint-Joris-Weert (L08_098) | Beauvechain (KMI) | Ukkel (KMI) |
| | | | Braine l'Alleud (KMI) | Zaventem (KMI) |
| | | | Chastre (KMI) | |
| | | | Marbais (KMI) | |
| | | | Ukkel (KMI) | |
| | | | Zaventem (KMI) | |
| Grote Nete | 1985 - 2017 | Hulshout (gnt05a-1066) | Kleine-Brogel (KMI) | Kleine-Brogel (KMI) |
| | | | Lommel (KMI) | Zaventem (KMI) |
| | | | Diest (KMI) | |

In addition, the model requires data on soil type, topography and land cover. Topographic information is derived from the SRTM 1 Arc-second Global version 3 dataset. Land cover information is derived from the 100-metre spatial resolution CORINE land cover maps of 1990, 2000, 2006 and 2012, which are simplified to five land cover classes (table A2) (EEA,

1995). The land cover at a certain time step is assumed to be equal to one of the four land cover maps, situated closest in time. The hydrological properties of the soil are derived from the European Soil Hydrogrids database. This database provides a wide variety of hydrological properties at seven distinct depths in the upper 2.5 metres of the soil profile with a spatial resolution of 250 metres (Tóth et al., 2017). The field capacity, wilting point and water holding capacity used in this study are calculated as an integration from the soil surface to a depth of 2.5 metres. For each of the simplified land cover classes, a specific land cover

coefficient value ($CROPF$) is used, which is based on independent evapotranspiration data (table A2) (Gellens-Meulenberghs and Gellens, 1992; Notebaert et al., 2011). All maps are resampled to a 50-metre resolution using bilinear interpolation.



**Table A2: Reclassification of the CORINE land cover classes used in the hydrological simulations and land cover coefficient values (CROPF) for the simplified land cover classes, used in the evapotranspiration calculations.**

| New land cover class | CROPF-value | CORINE Code | CORINE land cover class |
|---|---|---|---|
| Built-up area | 0.995 | 111 | Continuous urban fabric |
| | | 112 | Discontinuous urban fabric |
| | | 121 | Industrial or commercial units |
| | | 122 | Road and rail networks and associated land |
| Arable land/bare soil | 1.013 | 131 | Mineral extraction sites |
| | | 132 | Dump sites |
| | | 133 | Construction sites |
| | | 211 | Non-irrigated arable land |
| | | 331 | Beaches, dunes, sands |
| | | 333 | Sparsely vegetated areas |
| Pasture/grassland | 1.013 | 124 | Airports |
| | | 142 | Sport and leisure facilities |
| | | 222 | Fruit trees and berry plantations |
| | | 231 | Pastures |
| | | 242 | Complex cultivation patterns |
| | | 243 | Land principally occupied by agriculture, with significant areas of natural vegetation |
| | | 321 | Natural grasslands |
| | | 322 | Moors and heathlands |
| | | 412 | Peat bogs |
| Forest | 1.129 | 141 | Green urban areas |
| | | 311 | Broad-leaved forest |
| | | 312 | Coniferous forest |
| | | 313 | Mixed forest |
| | | 324 | Transitional woodland-shrub |
| Open water | 1.266 | 411 | Inland marshes |
| | | 511 | Water courses |
| | | 512 | Water bodies |

To calculate the amount of overland flow during rainfall events, curve numbers were assigned to each unique combination of the simplified land cover classes and hydrological soil groups (Soil Conservation Service, 1986; Suphunvorranop, 1985). The





texture classes of the Belgian soil map are reclassified into five hydrological soil groups (table A3). For locations in the landscape for which the soil texture was not identified, the dominant hydrological soil group in the river basin was assumed. As most open waters in both river basins are artificial and often used for rainwater buffering, they are not assumed to contribute

to the basin runoff and thus have a curve number value of 0.

**Table A3: Curve number values used for each combination of a hydrological soil group and a simplified land cover class. For each hydrological soil group, the corresponding texture classes according to the Belgian soil map are listed.**

| Hydrological Soil Group | Texture class according to the Belgian Soil Map | Simplified land cover class | Curve Number |
|---|---|---|---|
| A | P, S, Z, L-P-Z, P-Z, X, S-Z, A-Z (Sandy loam to sand) | Built-up area | 77 |
| | | Arable land/bare soil | 72 |
| | | Pasture/grassland | 39 |
| | | Forest | 25 |
| | | Open water | 0 |
| B | A, E, G, L, U-L, U-L-S, A-G, A-L, E-Z (Light clay to loam) | Built-up area | 85 |
| | | Arable land/bare soil | 81 |
| | | Pasture/grassland | 61 |
| | | Forest | 55 |
| | | Open water | 0 |
| D | U (Heavy clay) | Built-up area | 92 |
| | | Arable land/bare soil | 91 |
| | | Pasture/grassland | 80 |
| | | Forest | 77 |
| | | Open water | 0 |
| P | V (Peat) | Open water | 0 |
| | | All other land cover classes | 85 |
| W | Open water | All land cover classes | 0 |

The calibration procedure resulted in a model efficiency of 0.348 for the Dijle river and 0.422 for the Grote Nete for the daily discharge time series (table A4). Overall, the calibrated model versions are able to reproduce the mean discharge over the

calibration period relatively well with a simulated and observed mean discharge of 4.90 and 4.83 cubic metres per second for the Dijle river and 5.01 and 4.87 cubic metres per second for the Grote Nete river. Overall, significant differences occur between the observed and simulated discharge time series with mean relative error values of 23.1 percent for the Dijle river and 38.1 percent for the Grote Nete river.





**Table A4: Best-fitting parameter values, model efficiency and mean relative error (%) for the calibrated parameters of the STREAM model.**

| Calibration parameter | Dijle | Grote Nete |
|---|---|---|
| $P_{cal}$ | 2.952 | 3.257 |
| $S_{cal}$ (m day$^{-1}$ %$^{-1}$) | $3.866*10^{-4}$ | $1.057*10^{-3}$ |
| $G_{cal}$ (day$^{-1}$ %$^{-1}$) | 0.157 | 0.250 |
| Nash-Sutcliffe model efficiency | 0.348 | 0.422 |
| Mean relative error (%) | 0.231 | 0.381 |

Overall, the model efficiency for the best-fitting parameter combination is rather limited, especially for the Dijle basin, which also results in high mean relative errors (table A4) . It is however difficult to set a threshold value for the Nash-Sutcliffe model efficiency to determine when the model performance is sufficient, since that value is highly dependent on the model application (Beven, 2006). A model efficiency above 0 should be taken as a minimum since it indicates that the model predicts better than the mean value over the entire time series. In the literature, model efficiency values between 0.36 and 0.8 are mentioned as threshold values for acceptable model performance for various hydrological models and spatial and temporal scales (Knoben et al., 2019; Moriasi et al., 2007; Ritter and Muñoz-Carpena, 2013). The model efficiency values obtained here for the Dijle and Grote Nete rivers fall within the lower part of this range (table A4). However, the use of a model with three reservoirs limits the hydrology to a quick (runoff), intermediate (soil throughflow) and slow (groundwater flow) response to precipitation events, which are determined by the calibrated coefficients, limiting the freedom of the model. Additionally, the calibration procedure is based on the simulated and measured discharge time series. However, the measuring stations on the Dijle and Grote Nete rivers, operated by the VMM do not measure river discharge directly. Water stages are recorded and converted to discharges using empirical equations, which are updated once every few years. Given the use of a single conversion equation under varying conditions regarding channel vegetation and a changing cross-sectional area, it can be expected that the discharge value reported by the VMM deviates from the true river discharge. The effect of this error on the calibration procedure is difficult to quantify and as such, the discharge time series as provided by the VMM are assumed to be representative.

### A1.2    STREAM simulations for scenario analysis

To simulate the basin discharge under different climate and land cover scenarios, the land cover fractions must be allocated to provide land cover maps. However, it is unclear to what extent the spatial arrangement of the different land cover classes impacts the resulting basin hydrograph. A recent study on the Dijle river used both pollen-based vegetation reconstructions and sediment delivery modelling to come up with realistic land cover maps for six archaeological periods (Neolithic period, Bronze age, Iron age, Roman age, Early Medieval period and Late Medieval period). A total of almost 63,000 land cover maps were produced with vegetation fractions matching the pollen-based reconstructions but with varying land cover patterns. For each land cover map, the annual hillslope sediment delivery was modelled using the WATEM/SEDEM model. By comparing





the model output with the reconstructed geomorphic record for the Dijle river, land cover patterns with unrealistic sediment dynamics could be ruled out (De Brue, 2016). Here, for each archaeological period studied by De Brue, the land cover pattern was selected which matches best both the pollen-based vegetation and the geomorphic sediment delivery reconstructions (fig. A3 and table A5). These land cover maps can be assumed to be realistic patterns for each of the archaeological periods. For

each of these six periods, two hydrological simulations were run using the calibrated STREAM model for the Dijle basin over a period of 100 years. One with the realistic land cover pattern and one with a land cover map consisting of identical vegetation fractions but with random allocation. To rule out any climate effects, for all periods the simulations were run with the same climate scenario, namely the mean conditions for the Holocene. Differences between the two simulated hydrographs at the outlet of the basin can be used to identify the effect of land cover allocation on the hydrological simulations.






**Figure A3: Land cover maps for six archaeological periods (Neolithic period, Bronze age, Iron age, Roman age, Early Medieval period and Late Medieval period/Modern period), matching best the reconstructed pollen-based vegetation fractions and floodplain sediment delivery model for the Dijle basin.**





**Table A5: Vegetation fractions of the land cover maps of the Dijle catchment, used for the different archaeological periods.**

| Period | Forest | Grassland / short vegetation | Cropland / bare soil |
|---|---|---|---|
| Neolithic period | 86% | 7% | 7% |
| Bronze age | 74% | 13% | 13% |
| Iron age | 69% | 3% | 28% |
| Roman age | 66% | 17% | 17% |
| Early Medieval | 29% | 23% | 48% |
| Late Medieval/Modern | 1% | 10% | 89% |

The results indicate that the effect of the land cover allocation on the simulated discharges is rather small, as reflected in the high values for the coefficients of determination (fig. A4). However, for the archaeological periods in which the land cover is dominated by one land cover type, such as forest for the Neolithic period or cropland for the Late Medieval/Modern period, the difference between realistic and random allocation is clearly smaller than for periods during which the land cover was more

diversified such as during the Iron age and Roman age. Given these results, the land cover in the scenario analysis was allocated randomly.

**Figure A4: Scatterplots of the simulated discharge (m³ s⁻¹) for the realistic and randomly allocated land cover maps for each of the six archaeological periods (with indication of the coefficient of determination).**

### A1.3    Applying the STREAM model to the Dijle and Grote Nete river basins

The river basin hydrology was simulated over a period of 100 years for the Dijle and Grote Nete basins for each combination of the climate and land cover scenarios. The differences between the two contrasting river basins are for most hydrological fluxes larger than the range over all scenario combinations, demonstrating that the inherent differences in the basin hydrology between the Dijle and Grote Nete are more pronounced than variability due to the climate and land cover scenarios (fig. A5). While the different climate and land cover scenarios result in variability in the mean annual precipitation and actual evapotranspiration, the absolute values of the recharge rates remain stable with minimal variation over the different scenarios.



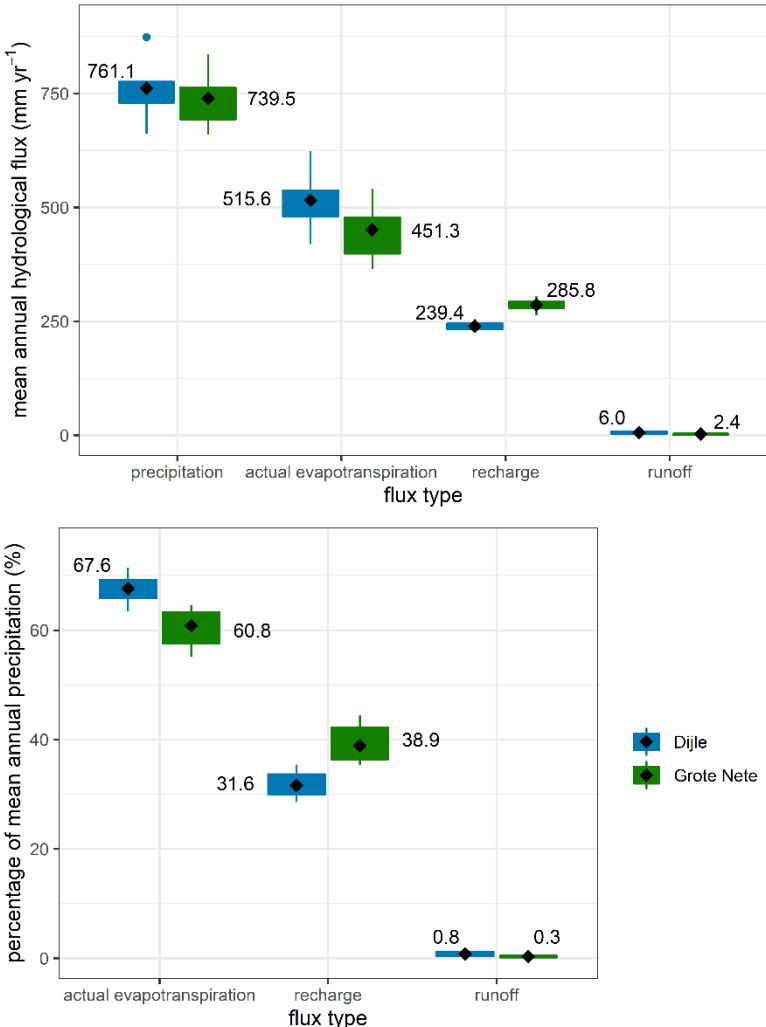

**Figure A5: Top: Mean annual hydrological fluxes (mm yr$^{-1}$) over the Dijle and Grote Nete basins for each combination of climate and land cover scenario (n = 30). Bottom: Relative mean annual hydrological fluxes as a percentage of the total annual precipitation over the Dijle and Grote Nete basins. The mean value is indicated by a black diamond. The numbers correspond to the mean value over all scenario combinations. The box indicates all values within the 25th to 75th percentile range and the whiskers represent all values within the range from the 25th percentile – 1.5*interquartile range to the 75th percentile + 1.5*interquartile range. Coloured dots represent all other values outside this range.**

The STREAM simulations can be compared to other studies to check how realistic the modelled results are. For the Grote Nete basin under current conditions, Batelaan and De Smedt calculate a mean recharge rate of 282 millimetres per year using the WetSpass model, which matches well with the mean value of 286 millimetres per year found here using the STREAM model (fig. A5) (Batelaan and De Smedt, 2001). For the Flemish part of the Dijle basin, they find a mean recharge percentage of 26 percent under current conditions (De Smedt and Batelaan, 2003). This estimate is lower than the mean value of 31.6 percent over all scenarios, found in this study, but given the low recharge percentages of built-up land and the absence of that land cover category in all land cover scenarios used here, the results of both models seem to converge relatively well.





The fluctuations in mean monthly discharge throughout the year are much larger for the Grote Nete river than for the Dijle river (fig. A6). Differences between the climate scenarios are most pronounced during winter, with a delay in the maximal mean monthly discharge for scenarios 1 and 2, which can be related to the colder temperatures for these scenarios resulting

in a delayed snow melt season. The different land cover scenarios result in small differences in the mean monthly discharge, with higher flows for the land cover scenarios with little or no forest cover, which can be attributed to the lower evapotranspiration flux over cropland and grassland, compared to forest. Overall, changes in land cover affect the mean monthly discharge of the Dijle river more in comparison to the Grote Nete river.

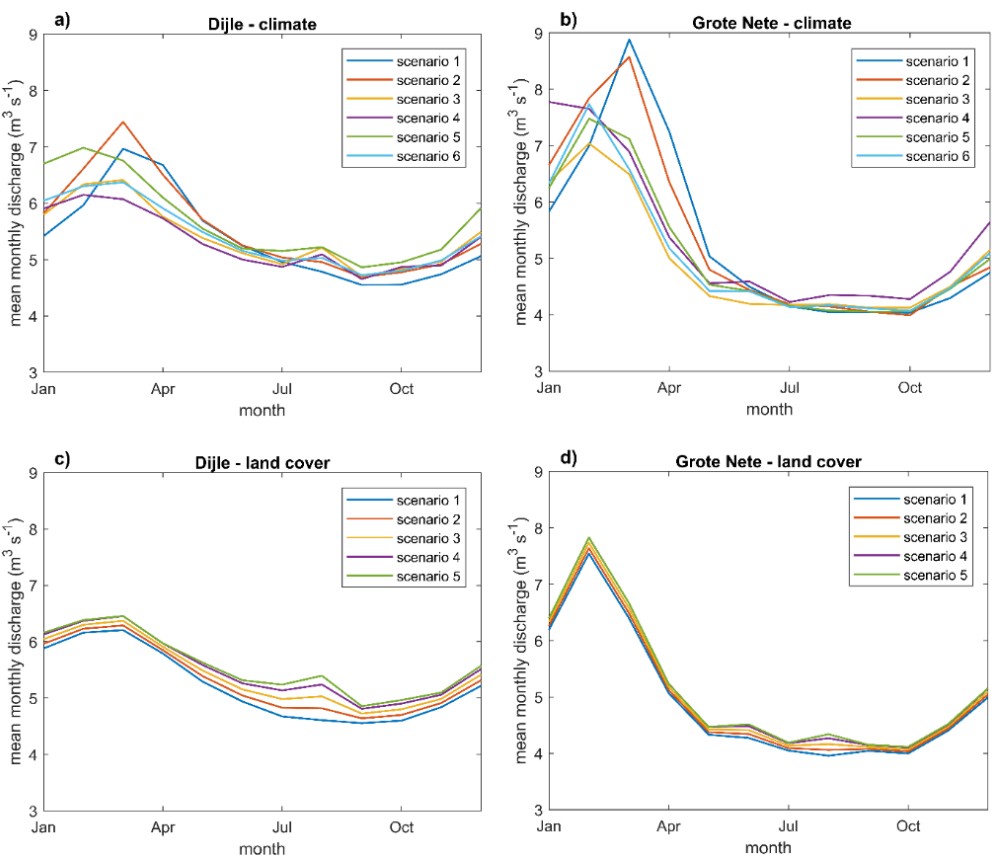

**Figure A6: Simulated mean monthly discharge (m³ s⁻¹) for the Dijle river at Korbeek-Dijle (a-c) and the Grote Nete river at Hulshout (b-d) under the different climate scenarios for land cover scenario 3 (a-b) and under the different land cover scenarios for climate scenario 6 (c-d).**

In addition to the mean discharges, recurrence intervals allow to study the changes in the frequency of peak flow events (fig. A7). The different land cover scenarios result in increasing discharge values with decreasing forest cover, although this effect

is larger for the Dijle river than for the Grote Nete river.





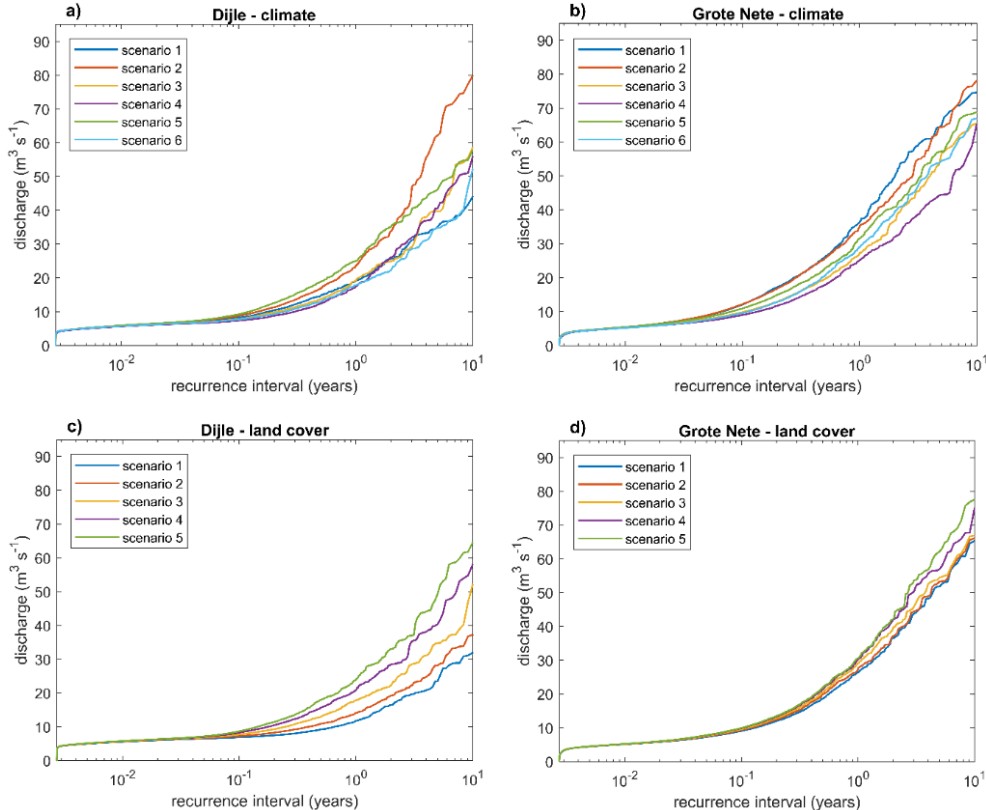

**Figure A7: Recurrence intervals (years) for the simulated daily discharge (m³ s⁻¹) for the Dijle river at Korbeek-Dijle (a-c) and the Grote Nete river at Hulshout (b-d) under the different climate scenarios for land cover scenario 3 (a-b) and land cover scenarios for climate scenario 6 (c-d).**

**A2      Modifications to the Digibog peatland model**

The local peatland model used in this study is based on the 2015 version of the Digibog peatland model, which was modified to be more applicable to an alluvial context. Firstly, the floodplain is assumed to consist of parallel river channels with peat bodies in between. The water table dynamics are calculated using Child's equation for elliptic bogs with the length of the ellipse (parallel to the river flow) being infinitely long (eq. (A1)).

$$\frac{dH}{dt} = \frac{U}{\theta_d} - \frac{K_{av} H^2}{L^2 \theta_d}$$   (A1)

With $H$ the water table elevation (m), $U$ the net rainfall (m), $\theta_d$ the drainable porosity of the peat (m³ m⁻³), $K_{av}$ the peat profile transmissivity (m² yr⁻¹) and $L$ the lateral extent of the peat bog, which equals half the spacing between the parallel drains (Morris et al., 2015).

Secondly, modifications were made to the calculation of the evapotranspiration fluxes. The annual potential evapotranspiration
rate, calculated using the Thornthwaite method, is subdivided in soil evaporation and plant transpiration based on the leaf area index (eq. (A2) and eq. (A3)) (Williams et al., 1983).





$$E_{soil} = E_{pot} \, e^{-0.4 \, LAI} \tag{A2}$$

$$E_{plant} = E_{pot} - E_{soil} \tag{A3}$$

With $E_{soil}$ the potential annual soil evaporation (m), $E_{plant}$ the potential annual plant transpiration (m), $E_{pot}$ the annual

potential evapotranspiration and $LAI$ the leaf area index (m² m⁻²).

The soil evaporation component of the total actual evapotranspiration rate is calculated based on the depth of the water table

(eq. (A4)). It is assumed that the soil evaporation rate is at its maximal value when the water table is located between the

surface and a specific depth, at which water supply for evaporation is not limited. Once the water table depth increases further,

the actual evaporation rate decreases with a linear trend until the depth for which water supply for evaporation reaches zero

(Price et al., 2003; Swinnen et al., 2019).

$$AET_{soil} = E_{soil} \, \frac{z_2 - wt}{z_2 - z_1}, \text{ for } z_1 \le wt \le z_2 \tag{A4}$$

With $AET_{soil}$ the actual soil evaporation rate (m yr⁻¹), $E_{soil}$ the potential annual soil evaporation rate (m yr⁻¹), $wt$ the water

table depth (m), $z_1$ the depth at which $AET_{soil}$ starts to decrease and $z_2$ below which $AET_{soil}$ becomes zero.

The actual plant transpiration rate is dependent on both the vegetation type and the water table depth. It is assumed that the

peat column is covered by a combination of tall vegetation (trees and shrubs) and short vegetation such as grasses, sedges and

mosses, each occupying between 0 and 100 percent of the surface, with the sum of both percentages being always 100 percent.

Similar to Heinemeyer et al., different vegetation classes are assumed a specific maximal rooting depth (Heinemeyer et al.,

2010). If the water table depth is at the maximal rooting depth, plants will still be able to transpire water due to capillary rise

of water into the rooted zone. As a result, the assumption is made that once the water table falls below the sum of the rooting

depth and the height of capillary rise ($z_{plant} + z_2$), the actual plant transpiration rate for this vegetation class falls to zero.

Additionally, the assumption is made that the actual plant transpiration rate is maximal when the water table depth is equal to

half this depth (($z_{plant} + z_2$)/2). Similar to Bauer et al., the rooting depth of the woody roots is assumed to be limited by the

water table depth. As a result, the maximal rooting depth cannot exceed the long-term mean water table depth over the past

ten years, with a minimum value of 50 centimetres (Bauer, 2004).

Different plant species respond differently to high water tables. Here, the assumption is made that plants which are typical for

wet conditions are able to grow and transpire water under high water table conditions, while the opposite is true for plants

which are typical for drier environments. These plants will suffer from the waterlogged conditions and stop transpiring water.

The preference of plant species with regard to moisture can be semi-quantified using the Ellenberg indicator value for moisture

(f-value), ranging from 1 to 12, with higher values indicating water-tolerant species. The actual plant transpiration rate when

the water table is at the surface is calculated based on the maximal actual transpiration rate and the Ellenberg value (eq. (A5)).

$$AET_{plant,waterlogged} = Eplant * \left(\frac{Ellenberg-f}{12}\right) \tag{A5}$$

With $AET_{plant,waterlogged}$ the actual plant transpiration rate (m yr⁻¹) under waterlogged conditions and $Ellenberg - f$ the

average moisture value for the vegetation class under consideration. This means that plant species with an Ellenberg f-value





of 12 (hydrophilic plant) will transpire at the maximal rate when the water table is located between the surface and half the
rooting depth, while plant species with a low Ellenberg f-value will suffer from high water tables and thus will have lower
rates of transpiration.

These assumptions result in defined actual transpiration rates for three water table depths (peat surface, half the maximal
rooting depth and maximal rooting depth). A continuous function expressing the relationship between the actual transpiration
rate and the water table depth is constructed using Piecewise Cubic Hermite Interpolating Polynomial splines based on the
three known points. As a result, an actual plant transpiration rate can be calculated for any given water table depth (fig. A8).
The total actual evapotranspiration rate thus consists of the sum of the actual soil evaporation rate and the plant transpiration
rates for both vegetation classes.

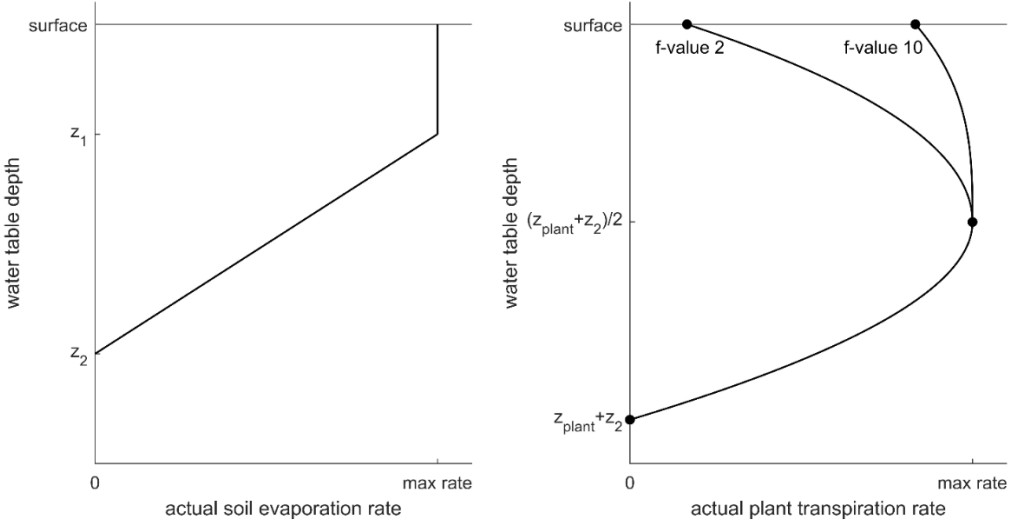

**Figure A8: Left: actual soil evaporation rate as a function of the water table depth. Right: actual plant transpiration rate as a**
**function of the water table depth and the Ellenberg indicator value for moisture (f-value). Examples are shown for f-values of 2 and**
**10.**

## A3      Field data

Given the idealised circumstances for which the model is developed, a detailed calibration and validation procedure is not
possible. However, several datasets, which are available for the studied case-studies, can be used to fine-tune specific model
parameters and constrain the uncertainty on the simulated results. A first source of information consists of the measured peat
thickness, derived from a dataset of 295 soil corings across the study areas (Swinnen et al., 2020). Since the Digibog model
does not incorporate sediment dynamics, the measured peat thickness values need to be corrected for compaction after being
buried by overlying mineral sediment. The compaction percentage is calculated using an empirical relationship, expressing the
percentage of thickness reduction as a function of the effective stress of the overlying sediment. Based on a study of Van





Asselen et al., compaction data from 14 buried floodplain peat deposits in the Cumberland Marshes (Central Canada) were

obtained, which were used to construct an empirical relationship (eq. (A6) and fig. A9) (Van Asselen et al., 2010).

$$C = 14.78 \sqrt[4]{2.801 \, \sigma'} \tag{A6}$$

With $C$ the peat compaction, defined as the reduction of the peat volume, expressed as a percentage of the original volume and

$\sigma'$ the effective stress exerted on the peat layer (kPa) (Van Asselen et al., 2010). This effective stress is calculated using

Terzaghi's equation, with, $\sigma$ the stress of the overlying sediment (kPa) and $u$ the pore water pressure (kPa) (eq. (A7) – (A9))

(Terzaghi, 1943).

$$\sigma' = \sigma - u \tag{A7}$$

$$\sigma = \rho_{sed} \, g \, h \tag{A8}$$

With $\rho_{sed}$ the saturated density of the overlying sediment (kg m$^{-3}$), $g$ the gravity constant (m s$^{-2}$) and $h$ the thickness of the

overlying sediment (m).

$$u = \rho_w \, g \, h_w \tag{A9}$$

With $\rho_w$ the density of the water (kg m$^{-3}$), $g$ the gravity constant (m s$^{-2}$) and $h_w$ the thickness of water column overlying the

peat layer (m). Given the high water table depths in floodplain environments, the effective overburden stress is calculated

under the assumption that the overlying sediments are water saturated.

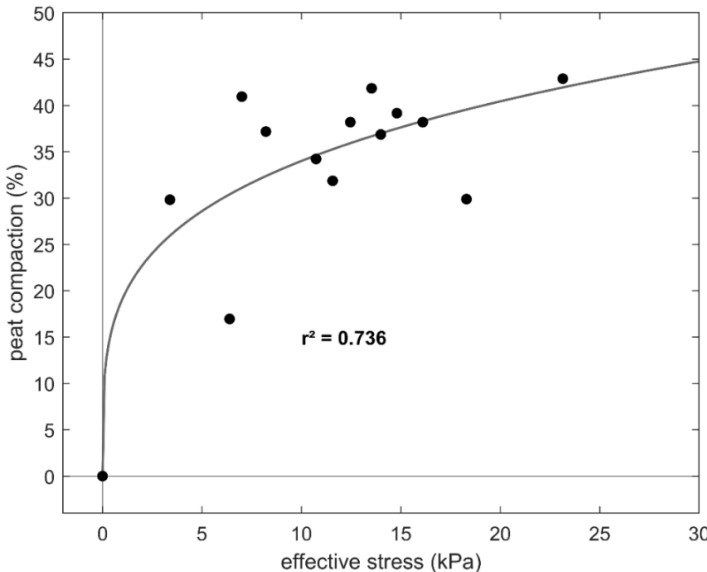


**Figure A9: Peat compaction (%) as a function of the effective stress exerted on the peat layer (kPa). The fitted fourth root function is based on a dataset for central Canadian floodplains (n = 14) (Van Asselen et al., 2010).**

The shape of equation A6 implies that a low effective stress of overburden sediment leads to compaction percentages of 20

percent or more, with the compaction rate decreasing with further increasing effective overburden stress. This can be explained

by the collapse of the physical structure of uncompacted peat at low effective stress values. Based on this relationship, the

compaction-corrected peat thickness can be calculated for all peat layers found in the soil coring dataset, which can be





compared with simulated peat thickness values. This results in a mean corrected peat thickness of 1.60 metres for the Dijle and 0.56 metres for the Grote Nete. Overall, the corrected thickness values range between 0.1 and 6.7 metres, although the higher thicknesses are only found in the Dijle river floodplains (fig. A10).

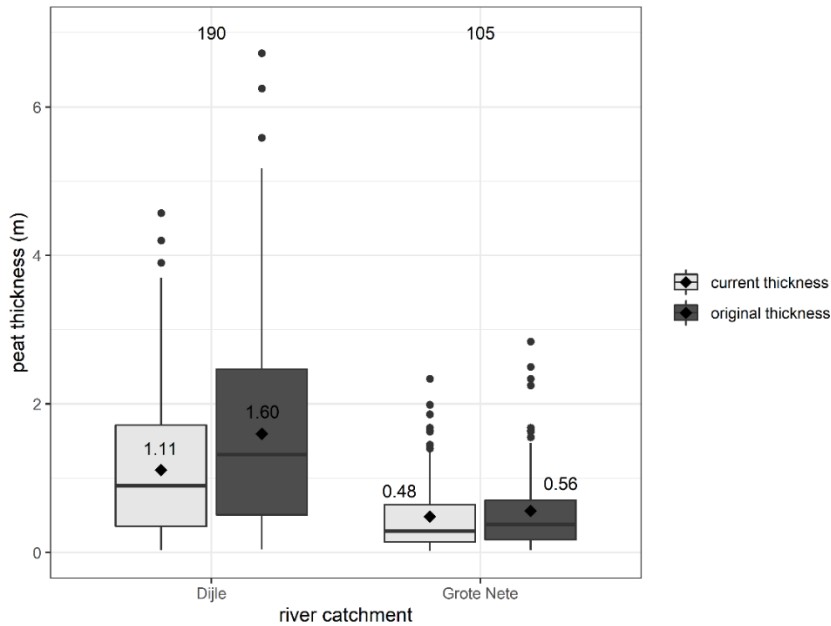


**Figure A10: Boxplots of the measured (*current thickness*) and compaction-corrected (*original thickness*) peat thickness for the Dijle and Grote Nete floodplains. The median and mean value are given by a horizontal line and black diamond. The values in the box indicate the mean value. The box represents all values within the 25th to 75th percentile range and the whiskers represent all values within the range from the 25th percentile – 1.5\*interquartile range to the 75th percentile + 1.5\*interquartile range. Black dots**
**represent all other values outside this range.**

In addition, available data on the dry bulk density and organic carbon content of the peat for each of the studied river systems (70 samples) can be used to calculate representative dry bulk density values for peat, correcting for both compaction due to burial and the presence of mineral sediment (eq. (A10)).

$$DBD_{cor} = DBD \ (1 - \ C)(\frac{\%OC}{0.58})$$ (A10)

With $DBD_{cor}$ the dry bulk density of the peat (g cm$^{-3}$), corrected for compaction and the presence of mineral particles, $DBD$ the mean measured dry bulk density for peat (g cm$^{-3}$), $C_{av}$ the compaction percentage for the region (loess belt and sand belt) and $\%OC$ the mean organic carbon content of floodplain peat for the region. Here, the organic matter present within the peat layer is assumed to have an organic carbon fraction of 0.58, which corresponds to the Van Bemmelen factor. Using equation A10, a mean dry bulk density and standard deviation of the mean for the organic matter fraction of peat of 0.11 ± 0.01 g cm$^{-3}$
was obtained for the Dijle river and 0.13 ± 0.01 g cm$^{-3}$ for the Grote Nete river. Based on these calculations, the standard value for the dry bulk density of peat used in model simulations is set to 0.12 g cm$^{-3}$.





## A4    Sensitivity analysis

For each of the model parameters considered in the sensitivity analysis, the simulated range, standard value and references are listed in the table below. For the model parameters related to the calculation of the evapotranspiration, not enough information

was found in the literature. As a result, these model parameters were not included in the sensitivity analysis and their value is kept at the standard value (table A6).

**Table A6: Overview of the parameters used in the parameter sensitivity test, listing the standard value and the range over which the parameter is varied.**

| Parameter | Standard value | Minimum value | Maximum value | References |
|---|---|---|---|---|
| **Peat properties** | | | | |
| Oxic decomposition rate (% $yr^{-1}$) | 4.2 | 0.93 | 6.33 | (Clymo, 1984; Kleinen et al., 2012; Lucchese et al., 2010; Malmer and Wallen, 2004; Wu, 2012; Yu et al., 2001b) |
| Anoxic decomposition rate (% $yr^{-1}$) | $2*10^{-2}$ | $4*10^{-3}$ | $2.36*10^{-2}$ | (Clymo, 1984; Clymo et al., 1998; Wu, 2012; Yu et al., 2001b) |
| $Q_{10}$-multiplier | 2.5 | 1.8 | 4.2 | (Chapman and Thurlow, 1998; Clymo, 1984; Stewart and Wheatly, 1990; Svensson, 1980; Wieder and Yavitt, 1994) |
| Peat dry bulk density (g $cm^{-3}$) | 0.12 | 0.07 | 0.45 | (Boelter and Blake, 1964; Chambers et al., 2011; Frolking et al., 2010; Granberg et al., 1999; Turunen et al., 2002) |
| Drainable porosity ($cm^{-3}$ $cm^{-3}$) | 0.30 | 0.2 | 0.55 | (Dasberg and Neuman, 1977; Kelly et al., 2014; Letts et al., 2000) |
| a-parameter hydraulic conductivity | $1.006*10^{-5}$ | $2*10^{-6}$ | $5*10^{-5}$ | (Morris et al., 2011) |
| b-parameter hydraulic conductivity | 8 | 5 | 11 | (Morris et al., 2011) |
| **Environmental parameters** | | | | |
| Mean annual temperature (°C) | 8.89 ± 0.72 | 4.5 | 12.5 | (Lawrimore et al., 2011; Mauri et al., 2015) |
| Mean annual precipitation (m) | 0.762 ± 0.114 | -75% | +75% | (Mauri et al., 2015; Peterson and Vose, 1997) |
| Lateral extent (m) | 100 | 10 | 500 | / |
| Vertical water flux (% of net annual rainfall) | 0 | -75 | 75 | / |
| **Vegetation properties** | | | | |
| Fraction open vegetation (%) | 50 | 0 | 100 | / |
| Mean Ellenberg moisture value (f-value) | 9 | 1 | 12 | (Ellenberg, 1974) |
| $z_1$ (depth where $AET_{soil}$ starts to decrease) (m) | 0.3 | / | / | (Wosten and Ritzema, 2001) |
| $z_2$ (capillary rise) (m) | 0.5 | / | / | (Price et al., 2003) |
| Minimum rooting depth trees (m) | 0.5 | / | / | (Heinemeyer et al., 2010) |
| Rooting depth open vegetation (m) | 0.5 | / | / | (Heinemeyer et al., 2010) |



## Author contributions

The conceptualization of this project was carried out by WS, NB and GV. WS carried out the modelling work and analysis. NB and GV supervised the research. The writing of the manuscript was carried out by WS, NB and GV.

## Acknowledgements

This research is funded by FWO (Research Foundation Flanders - applications G0A6317N and 1167019N) and FWO-SBO project *Future Floodplains* (application S003017N).

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
