# Peer review of "Modelling long-term alluvial peatland dynamics in temperate river floodplains"

_Biogeosciences, 2021_

## Author Comment (AC1)

**Author response to the interactive comment RC1 by Paul J. Morris**

In the text below, the authors respond to the comments given by Paul J. Morris. The comments of the referee are given as plain text, while the authors response is given in *italic*.

**Comments**

OVERVIEW
This is an interesting and largely well-written paper that describes the development and analysis of a new coupled model of channel dynamics and Holocene peat development in lowland riverine peatlands. The coupled model consists of the existing STREAM catchment model and the DigiBog model of peat development, which have been joined in thoughtful way to simulate what are arguably under-studied landscapes. The authors conclude that, unlike more commonly modelled ecosystems such as raised bogs, the Holocene development of riverine peatlands is likely to be more heavily influenced by channel dynamics than by internal feedbacks. The main contribution of the article is to provide the community with a new tool with which to study these ecosystems, which will surely be of interest to the journal's readership. The methods are detailed, and for the most part clear. I don't have any major concerns about the technical accuracy of the paper, but I do have some suggestions for improved presentation in some places.

SUBSTANTIVE COMMENTS
The methods are rather long, especially when one considers the extensive methodological detail in the supplementary material as well as the main article. This isn't a problem in itself; rather, I commend the authors for providing full and reproducible methodological details. However, the methods comprise multiple sequential steps of model development, which are introduced in a piecemeal fashion. There are the field data, their incorporation into the two models, algorithmic alterations to then two models, their coupling, the sensitivity analysis, and experimentation with an aggrading channel. Some of the alterations to DigiBog are quite extensive (productivity, water balance), but are rather hidden in the appendix – a short mention here would be appropriate. I for one would have benefitted from a short, crisp overview at the very beginning of the methods (perhaps just a few sentences, in plain language), that primes the reader for all the various methodological steps and how they fit together. Some, but not all, of this information is conveyed in the final paragraph of the otherwise well-written introduction. This paragraph starts off with a statement of aim, but then rather drifts. I recommend having a short statement of purpose at the end of the introduction, and a methodological summary at the start of the methods.

*We agree with the reviewer that this article indeed contains an extensive methodological section. To improve readability, we will add a short paragraph between the introduction and methodology, which outlines the structure of the article and gives a short overview of the methodology. A more detailed description will then be given in the following sections and in the appendices. As suggested by the reviewer, the final paragraph of the introduction will be rewritten to present the aim of the article more clearly.*

On line 151, we are told that the simulated peatland is infinitely long in the along-stream axis. This is probably a reasonable starting point for these kinds of simulations. However, the diagram in Figure 3a seems to show the peat deposits as quite short, fat ellipses in plan, far from infinitely long. Figure 3a is probably closer to the reality of the situation, but isn't a good representation of the idealised simulations. I recommend adjusting Fig 3a to show peatlands then are (infinitely-) long, thin rectangles in plan, not ellipses. Figure 3b is fine.

*The representation of the peat layers in figure 3a can indeed be somewhat misleading. The figure will be modified according to the comment given by the reviewer.*

On lines 184-5, we read that the simulated peatlands are not eroded by a laterally-meandering stream. I can't help but wonder if this is a realistic assumption for lowland riverine peatlands. Again, simulations with such a novel modelling framework have to start somewhere, and it makes sense to add complexity incrementally rather than all in one go. So I am not suggesting altering the model specification. Nonetheless, some consideration here of the limitations of this assumption would be appropriate. Basal radiocarbon dates can tell us about how old the peat can get before it is eroded away.

*That is indeed true. Peat erosion by channel migration was kept out of the simulations to avoid making the scenarios overly complex. On the other hand, the radiocarbon dates we collected over the past years indicate that in many floodplain locations the complete Holocene peat record is still present. As a result, we hypothesize that peat erosion by channel dynamics will have taken place at some locations, but that significant sections of the floodplains were able to develop peat continuously throughout the Holocene without peat removal by channel migration. A few sentences will be added to the manuscript to clarify why we chose for this approach.*

Figure 6 has an unusual presentation given that both temperature and precipitation are time series. I'm all for innovative graphics as long as they convey the intended meaning clearly, and I think it's useful to show these scenarios on climate space like this. However, I think it would also help to have two extra panels in the figure to display the two variables as time series for each of the two catchments, and to have numbered shaded bars to indicate the climate space of the six scenarios.

*Providing a graphical representation of the temporal evolution of the two climate variables can indeed be helpful to understand the Holocene climate evolution over the study area and the time periods corresponding with the selected scenarios. Two additional panels will be added to the figure displaying the temporal evolution of the mean annual temperature and precipitation, with indications of the selected climate scenarios.*

MINOR AND PRESENTATIONAL COMMENTS

Line 43: Here and throughout, the term "peat bogs" is somewhat ambiguous, and would be more clearly presented as "raised bogs".

*This will be changed in the new version of the manuscript to avoid confusion.*

Line 89: Here and throughout, I think "allows to" should be "allows us to".

*This will be changed in the new version of the manuscript.*

Lines 210 – 214: Good idea to experiment with these parameters. From what we know about previous versions of DigiBog, anything that affects the water budget that is retained by/lost from the model is likely to be highly influential upon peat accumulation and development.

*Thank you for the comment. The results of the sensitivity analysis indeed indicate the strong effect of the "hydrological setting" on the resulting peat thickness.*

Lines 216 – 221: I agree with this interpretation that very low oxic decay rates and well-preserved peat are unlikely to be able to retain much water, thereby stunting peat growth. Again, this is exactly what we see in previous versions of DigiBog.

Line 247: The calendric unit BCE is somewhat awkward. I take this to mean before current era, in other words equivalent to the Christian calendar. In which case, 10,005 BCE equates to 11,955 BP (where 0 BP is 1950 AD). Converting from unusual date formats quickly gets tangled. I recommend presenting these dates in a more familiar format. What's wrong with BP, which is very widely used in Quaternary/Holocene studies? Either way, please define the baseline (zero) year unambiguously.

*The term "BP" is indeed often used in Holocene and Quaternary studies to indicate time. However, "BP" and "cal BP" find their origin in the field of radiocarbon dating and in a strict interpretation, these terms are not applicable outside of this context. As such, in this article we use BP or cal BP for all dates related to radiocarbon dates and BC/BCE for all other dates (e.g. simulation years). This approach is based on a comment, published in Quaternary Science Reviews, which argues that the terms "BP" and "cal BP" should be exclusively used for radiocarbon dates (Anon, 2007; Wolff, 2007). The use of BC/BCE for calendar years is based on the author guidelines of Biogeosciences.*

*In this article, the terms BP/calBP are used when discussing the climate reconstructions or for dates derived from radiocarbon samples. These climate reconstructions were derived from a published dataset, which is based on radiocarbon dated pollen records and reports the results/dates in BP (Mauri et al., 2015). As a result, we took over the same unit.*

*A line will be added to the manuscript to clarify this choice.*

Lines 251 – 252: I think "selecting randomly daily times series" should be "randomly selecting daily time series".

*This will be changed in the new version of the manuscript.*

Line 271: Now BP is being used for dates. Suggest using a consistent calendar throughout.

*See comment above about the use of BP/cal BP.*

Figure 8b: Suggests that the river would erode the peat. See earlier comment about acknowledging the assumption that it does not.

*Lateral erosion is indeed not taken into account. See comment above.*

**References**

*Anon: The use of time units in Quaternary Science Reviews, Quat. Sci. Rev., 26(9), 1193, doi:https://doi.org/10.1016/j.quascirev.2007.04.002, 2007.*

*Mauri, A., Davis, B. A. S., Collins, P. M. and Kaplan, J. O.: The climate of Europe during the Holocene: A gridded pollen-based reconstruction and its multi-proxy evaluation, Quat. Sci. Rev., 112, 109–127, doi:10.1016/j.quascirev.2015.01.013, 2015.*

*Wolff, E. W.: When is the "present"?, Quat. Sc*i. Rev., 26(25–28), 3023–3024, doi:10.1016/j.quascirev.2007.10.008, 2007.

---

## Author Comment (AC2)

**Author response to the interactive comment RC2 by an anonymous referee**

In the text below, the authors respond to the comments given by an anonymous reviewer. The comments of the referee are given as plain text, while the authors response is given in *italic*.

**Comments**

The authors investigate the use of a newly coupled accumulation model (Digibog) and a simple hydrological model (Stream) to simulate alluvial peatland development over Holocene timescales in relation to changes in both local and regional environmental conditions. A scenario-based approach was used to assess the sensitivity of alluvial peat growth to environmental changes under a wide range of settings. The results demonstrate that the alluvial peatland dynamics appear to be strongly determined by the setting and dynamics of the local river network, rather than by internal peatland dynamics or regional environmental changes.

This is an excellent paper. It is highly descriptive and well written. Technicalities about the model are well described. As already mentioned by the other reviewer, I believe that the authors should include an overview at the very beginning of the methods that primes the reader for all the various methodological steps and how they fit together. Apart from that and minor corrections (listed below) I believe that the article should be accepted subject to technical corrections.

*Thank you for the comments. An additional section will be added between the introduction and the methodology sections, which provides a short overview of the workflow and the used methodology. We hope that the addition of this paragraph improves the readability of the manuscript and provides a short but clear overview of the article.*

On line 229 it is written that "This indicates that the increased biomass productivity due to higher temperatures does not compensate the temperature effects". Do the authors mean "This indicates that the increased biomass productivity due to higher precipitation does not compensate the temperature effects"?

*The formulation of this sentence might indeed be somewhat confusing. A higher temperature has a positive effect on the biomass productivity, given the sufficient water supply in temperate floodplains. This has a positive effect on the peat accumulation rate. On the other hand does an increased temperature also lead to higher evapotranspiration rates and biomass decomposition rates, which negatively affect the peat accumulation rates. As the sensitivity analysis shows a decrease in peat thickness with rising temperatures, we conclude that the positive effect on the biomass productivity does not outweigh the negative effects on biomass decomposition and evapotranspiration. The sentence will be modified to convey this message more clearly.*

Figure 4 - Could you provide more information about what means scaled parameter? To my point of view, the x-axis should be graduated.

*The label on the x-axis ("scaled parameter") can indeed be somewhat confusing. Each parameter is varied over a specific range, which is mentioned in table A6. As such, the absolute values on the x-axis are different for each parameter. To present the figure more clearly, the label will be changed to "minimum value" and maximum value", with a reference to table A6 in the figure caption. This allows the reader to look up the minimum and maximum value of the simulated range for each of the parameters in table A6 and keeps the nomenclature consistent between figure 4 and table A6.*